# From Science to Dressing Room: Dietary Supplements for Elite Soccer Performance

**DOI:** 10.3390/jfmk10040408

**Published:** 2025-10-21

**Authors:** Tindaro Bongiovanni, Federico Genovesi, Christopher Carling, Gianpiero Greco, Ralf Jäger

**Affiliations:** 1Department of Biomedical and Neuromotor Sciences, University of Bologna, 40126 Bologna, Italy; tindaro.bongiovanni2@unibo.it; 2Manchester City Football Club, Player Health & Performance, Manchester M11 3FF, UK; federico.genovesi@mancity.com; 3Laboratoire Sport, Expertise and Performance INSEP, 75012 Paris, France; christopher.carling@gmail.com; 4Department of Translational Biomedicine and Neuroscience (DiBraiN), University of Study of Bari, 70124 Bari, Italy; 5Increnovo LLC, Whitefish Bay, WI 53217, USA; ralf.jaeger@increnovo.com

**Keywords:** soccer, ergogenic aids, recovery, performance, dietary supplements

## Abstract

**Purpose:** The aim of this review is to provide an overview of the effects of commonly used dietary supplements on soccer performance and to bridge the gap between scientific evidence and their practical application by practitioners working with elite soccer players. **Methods:** Relevant literature involving dietary supplement use in soccer players was identified through searches of PubMed, EMBASE, Scopus, and Web of Science. Additionally, insights were gathered from a cross-sectional online questionnaire completed by practitioners (nutritionists, physicians, sport scientists, strength and conditioning coaches, and heads of performance) working with first-division men’s teams across five European leagues. Eligible respondents were over 18 years old with >2 years of experience in elite sport. The 20-question survey, designed on Qualtrics and pilot-tested for content validity, covered practitioner background, beliefs about supplementation, and real-world practices. The study was approved by the Ethical Independent Committee in Genoa, Italy (Ref. 2020/12). **Results:** Among performance-enhancing supplements, caffeine has been shown to improve endurance, sprint performance, power, and cognitive function, while creatine consistently enhances short-duration, high-intensity efforts. Beta-alanine and sodium bicarbonate help reduce the buildup of acidity in muscles during repeated high-intensity exercise, supporting repeated sprint performance. For hydration and endurance support, dietary nitrates improve blood flow and oxygen delivery to muscles, and glycerol enhances fluid retention in hot environments and during compressed match schedules, where players compete in multiple matches within a short recovery window. Regarding recovery aids, protein and tart cherry supplementation have been shown to accelerate recovery, reduce muscle damage, and support training adaptations. Field insights revealed that creatine and caffeine were widely adopted by practitioners (>90%), with protein powders also commonly recommended (>80%). In contrast, beta-alanine, tart cherry, and dietary nitrates were only partially integrated into daily practice (30%, 32%, and 48.5%, respectively), while sodium bicarbonate (24%) and glycerol (10.5%) were used by a minority. **Conclusions:** Although scientific evidence provides a strong foundation for the efficacy of dietary supplements, their translation into elite soccer practice is shaped by a range of practical factors, including cultural resistance, taste preferences, gastrointestinal side effects, established team routines, and individual player preferences. These findings highlight the importance of targeted education for players and staff, individualized supplementation plans, and close collaboration between nutritionists, coaches, and medical teams. However, our survey did not directly assess reasons for non-implementation. In addition to practical barriers reported by practitioners, unfamiliarity with current evidence likely contributes to this evidence–practice gap.

## 1. Introduction

Dietary supplements encompass a diverse range of substances, and their definition often varies depending on the regulatory, clinical, or performance context. A recent consensus by the International Olympic Committee (IOC) defines a dietary supplement as “a food, food component, nutrient, or non-food compound that is intentionally consumed in addition to the habitual diet to confer a specific health and/or performance benefit” [1]. In the context of this review, dietary supplements are considered as products intentionally consumed in addition to the habitual diet and typically delivered in concentrated forms such as capsules, tablets, or powders. Protein powders, for example, are included because they are marketed, consumed, and regulated as dietary supplements. In contrast, carbohydrates, while critical for fueling performance, are most often provided as part of foods or beverages rather than as discrete supplement products and therefore fall outside the scope of this analysis.

Supplements are widely used in elite sport to prevent nutritional deficiencies, accelerate recovery from intense physical exertion, and enhance performance [2]. In elite soccer, dietary supplements are commonly incorporated into nutritional strategies designed to support players’ performance and recovery [3]. Data show that between 47.8% and 93.7% of soccer players use dietary supplements to maintain health and improve physical performance [4,5,6]. Surveys also indicate that products such as whey protein, caffeine, and creatine remain among the most frequently consumed dietary supplements by elite players [4]. Despite this high prevalence, the scientific literature on their efficacy for achieving peak soccer performance remains limited, partly due to the challenges of conducting research with professional athletes who face compressed competitive schedules [2].

“Soccer performance” is a multifaceted construct that integrates physical, technical, tactical, and cognitive components. Peak performance, often described as the optimal state of physical, psychological, and tactical readiness, emerges from the complex interaction of these factors [7]. Physically, players must repeatedly perform explosive activities such as sprints, jumps, accelerations, decelerations, and rapid changes of direction interspersed with lower-intensity recovery periods [8]. High-intensity running distance during matches is a well-established indicator of performance [9]. Beyond physical abilities, cognitive skills such as information processing, decision-making, and anticipation allow players to respond with speed and accuracy to rapidly changing game contexts [10,11]. These demands underscore the need for nutritional strategies that support not only physical output but also cognitive function and recovery.

Coaches, sports nutritionists, sports scientists, and physicians play an important role in understanding and applying knowledge about dietary supplements, nutrient timing, and athletic performance, although levels of expertise and familiarity with current evidence can vary considerably across these groups [7,12,13]. For example, Arsène Wenger emphasized that nutrition should support players’ health and overall wellness, help optimize performance by ensuring they are in prime physical condition for each match with optimal body composition, facilitate rapid recovery after matches or intense training sessions, and provide adequate fuel to sustain high-intensity effort for 90 min or more [14]. The concept of optimization involves targeted interventions, such as carbohydrate periodization, that align nutrient availability with physiological demands to enhance adaptation and performance [15,16,17,18]. A key aspect of nutritional strategy is understanding how different approaches, including, but not limited to, supplementation, can influence sport-specific performance.

Although many supplements are marketed for performance enhancement, their evidence base varies considerably, and not all have demonstrated efficacy in soccer-specific contexts [13,19,20]. Evidence-based frameworks, such as the Australian Institute of Sport’s ABCD Classification and the International Society of Sports Nutrition’s position stands, help guide supplement selection by categorizing ingredients according to scientific support and safety [1,21]. Caffeine, beta-alanine, creatine, sodium bicarbonate, dietary nitrates, and glycerol are among the few supplements with consistent evidence supporting their ergogenic effects, while protein, essential amino acids, and tart cherry extracts are supported for recovery [22]. It is also important to recognize that physiological responses to some supplements, such as dietary nitrate, can vary by training status, with effects often observed in recreational athletes but not in well-trained players [20].

Despite these advances, an important knowledge gap remains: how is this scientific evidence translated into daily practice within elite soccer environments? Specifically, how do practitioners integrate evidence-based supplementation strategies into individualized performance programs for professional players? This narrative review aims to address these questions by providing a comprehensive overview of the effects of dietary supplements on soccer performance, based exclusively on studies conducted in soccer players, and by examining how practitioners implement supplementation strategies in the context of elite professional sport [23]. It should be noted that this definition of professional soccer players is provided for contextual understanding only and was not used as an inclusion or exclusion criterion in the literature search.

## 2. Materials and Methods

### 2.1. Search Strategy and Study Selection

Literature searches were conducted between March 2024 and March 2025 using the following databases: PubMed, EMBASE, Scopus, and Web of Science. Keywords such as “performance supplements” and “soccer” were used, individually combined with terms including “supplementation,” “athletes,” “ergogenic,” “peak performance,” “nutritional strategies,” and “ergogenic aids.” To ensure sport-specific relevance and ecological validity, this review was deliberately limited to studies conducted exclusively in soccer players. Articles were eligible for inclusion if they specifically investigated the use, prevalence, or effects of dietary supplements among soccer players, or if they provided theoretical or practical perspectives relevant to performance, recovery, or health in this population. Mixed-sport studies were excluded unless results were reported separately for soccer players. Although a significantly larger body of literature exists on these supplements in other sports, we did not include such studies because differences in physiological demands, positional roles, and competitive structures limit their direct applicability to soccer. Both observational and interventional studies, as well as narrative and systematic reviews, were considered to ensure a comprehensive and balanced overview. Only articles published in English were included. Exclusion criteria comprised duplicate reports and studies lacking methodological clarity. The selection process prioritized relevance, impact, and diversity of perspectives, aiming to capture not only empirical findings but also conceptual discussions surrounding the role of dietary supplementation in soccer performance. This flexible yet transparent approach follows established best practices for narrative reviews [24]. A narrative review was selected as the most appropriate format because the goal was to highlight key issues, identify existing research gaps, and illustrate both the current state of knowledge and areas where evidence remains limited. The specific literature cited throughout this review was selected based on its representativeness of broader trends and its contribution to understanding the evolving role of dietary supplementation in soccer.

### 2.2. Practitioner Recruitment and Survey Methodology

Fifty practitioners working with elite soccer players were recruited via personal messaging applications (e.g., WhatsApp), email, and social media platforms (e.g., Instagram, Facebook, and LinkedIn) through the professional networks of the research team. Participants represented first-team squads from the five highest-ranked European soccer leagues, the Italian Serie A, English Premier League, Spanish La Liga, German Bundesliga, and French Ligue 1, providing a broad and representative overview of current practices at the highest level of the sport.

Eligibility criteria included being over 18 years of age, having more than two years of experience working in an elite sporting environment, and currently serving as a physician, nutritionist, head of performance, sport scientist, or strength and conditioning coach with an elite-level men’s team competing in the highest professional division. The study was approved by the Ethical Independent Committee for Clinical, Non-Pharmacological Investigation in Genoa, Italy, and conducted in accordance with the ethical principles of the 1964 Declaration of Helsinki (Ref. 2020/12).

A cross-sectional online survey was designed to investigate practitioners’ perceptions, attitudes, and practices regarding dietary supplement use in elite soccer. The survey was developed in English using the Qualtrics CORE XM platform and underwent pilot testing (n = 5) to ensure clarity and content validity. Pilot responses were excluded from the final analysis. The finalized questionnaire, consisting of 20 questions, was distributed between March and July 2024, and data collection continued until December 2024.

The questionnaire was developed based on the subject-matter expertise of the authors (T.B., F.G., and C.C.) and refined through consultation with academic peers and football practitioners. It was structured into three main sections: 1. Participant Background Information: This section gathered demographic and professional data, including age, league affiliation, years of experience, highest level of education, and current role within the club. 2. Perceptions and Attitudes Toward Dietary Supplements: Participants rated their agreement with statements on knowledge, interest, player demand, and perceived safety and efficacy using a 5-point Likert scale (1 = strongly disagree to 5 = strongly agree). Example items included: “I am knowledgeable about dietary supplements,” “There is a high demand for dietary supplements among players,” and “Dietary supplements are safe and effective.” 3. Real-World Practices and Implementation: This section explored the types of supplements used or recommended (e.g., creatine, caffeine, dietary nitrates, beta-alanine, protein), as well as details on formulation, dosage, and timing strategies. Participants were asked both closed-ended questions (e.g., “Do you use creatine?”) and open-ended follow-ups (e.g., “If yes, describe the formulation and dosing protocol”).

Responses from questions with preset answer formats (e.g., multiple-choice or 5-point Likert scale items) were coded and processed using a standardized Microsoft Excel spreadsheet for subsequent analysis. The combination of Likert-scale, multiple-choice, and open-ended items enabled both quantitative assessment of supplementation patterns and qualitative insight into practitioner decision-making.

## 3. Results

### 3.1. Caffeine

Caffeine is a central nervous system stimulant that functions primarily as an adenosine receptor antagonist, reducing perceived exertion and enabling athletes to sustain higher intensities at a given effort level [25,26]. Pre-exercise caffeine ingestion can improve cognition, attention, vigilance, and exercise performance, although the magnitude of these effects depends on factors such as exercise type, duration, and dosage [27]. In elite soccer players, doses of 3–6 mg/kg body mass taken approximately 60 min before activity have consistently been shown to enhance performance [22,27]. While this dosage range is well established, it remains unclear whether lower doses (e.g., 3 mg/kg) provide similar ergogenic benefits to higher doses (e.g., 6 mg/kg). Importantly, doses exceeding ~6 mg/kg have not consistently demonstrated additional ergogenic effects and may increase the risk of adverse outcomes, such as gastrointestinal distress, anxiety, or impaired sleep, potentially negating any performance benefits.

Foskett et al. (2009) [28]. reported that acute caffeine ingestion (6 mg/kg), consumed 60 min before a 90-min soccer-specific simulation, significantly improved passing accuracy and jump height. Enhanced passing precision can improve overall team coordination, facilitate successful attacking transitions, and increase the likelihood of creating goal-scoring opportunities. Likewise, improved jumping performance not only strengthens a central defender’s aerial presence but also benefits forwards and midfielders during set pieces, contested balls, and transitional phases of play. Guerra Jr. et al. (2018) [29] demonstrated that caffeine ingestion (5 mg/kg) potentiated jump performance following acute plyometric and sled-towing exercises. Similarly, Nakamura et al. (2020) [30] observed that caffeine (3 mg/kg) improved intermittent sprint performance in highly trained male players. These improvements are relevant in match scenarios where repeated sprints and overlapping runs, such as those performed by full-backs or wing-backs, are critical to goal-scoring opportunities.

Alternative delivery methods, such as caffeinated gum, have also shown efficacy. Ranchordas et al. (2018) [31] reported that chewing gum containing 200 mg of caffeine for five minutes before testing enhanced countermovement jump, sprint performance, and Yo-Yo Intermittent Recovery Test outcomes compared with placebo. The Yo-Yo Intermittent Recovery Test is a field-based assessment designed to evaluate an athlete’s ability to repeatedly perform intense exercise bouts with brief recovery periods, a key physical demand in soccer. The test consists of repeated 20 m shuttle runs at progressively increasing speeds, interspersed with short recovery intervals, and is widely used to assess players’ aerobic and anaerobic capacity, as well as their ability to sustain high-intensity efforts throughout a match. Yildirim et al. (2023) [32] found that a similar dose (200 mg) improved quadriceps strength, ball-kicking speed, and jump performance compared with a lower dose (100 mg) or placebo. In female soccer players, caffeine (3 mg/kg) provided as an energy drink increased countermovement jump height, peak sprint speed, total running distance, and high-intensity activity during simulated match play (Lara et al., 2014) [33]. A systematic review by Abreu et al. (2023) [2] concluded that caffeine, consumed alone or with carbohydrates in beverages such as Red Bull or Fure, enhanced repeated sprint ability, maximal jump height, and total running distance in elite players. It should be noted that many of the studies included in the review used commercial energy drinks that contained additional ingredients such as taurine, B-vitamins, or herbal extracts. As a result, the observed effects cannot be attributed exclusively to caffeine, although the caffeine content likely played a major role in the ergogenic response. It is important to note, however, that most of these findings are derived from controlled or simulated practice settings designed to replicate match demands. Although these studies provide valuable insights, further research conducted during actual competitive match play is needed to confirm whether these performance enhancements translate directly to in-game scenarios.

Responses to caffeine vary considerably between individuals. Caffeine is a hydrophobic molecule that is rapidly absorbed and widely distributed throughout the body, with peak plasma concentrations occurring 30–120 min after ingestion. Caffeine metabolism and absorption kinetics can be influenced by both the total dose ingested and the form of administration; higher doses tend to prolong half-life, while anhydrous preparations generally produce faster absorption and higher peak plasma concentrations than coffee or energy drinks, in which other bioactive compounds can modulate these effects. Absorption is generally unaffected by age, sex, or health status, with ~99% of caffeine absorbed within 45 min of ingestion. However, metabolic rate can vary considerably due to genetic and environmental factors, including hormonal status such as the use of oral contraceptives, resulting in ‘fast’ and ‘slow’ metabolizers (Rodak et al., 2021) [34]. Additionally, individuals who are not accustomed to caffeine use or who are particularly sensitive to its effects may experience adverse reactions such as nervousness, restlessness, headaches, gastrointestinal discomfort, or insomnia even at moderate doses. Those who consume high doses from multiple sources (e.g., coffee, anhydrous caffeine, energy drinks, gels, or gum) are at even greater risk of these side effects, which can include more pronounced sleep disruption, anxiety, and digestive issues (Nédélec et al., 2015) [35].

Sleep is a crucial component of post-exercise recovery and fatigue reduction [36]. Sleep deprivation increases cortisol levels [37], impairs glycogen resynthesis [38], dysregulates appetite, and alters energy expenditure [39]. Because acute caffeine ingestion can delay sleep onset and decrease sleep quality, caution is warranted when dosing in the evening, particularly among caffeine-sensitive individuals [40]. Late-evening matches, now common due to broadcasting schedules, exacerbate this issue, with many players reporting difficulty sleeping until early morning hours following caffeine consumption. This reinforces the need for individualized dosing strategies to minimize sleep disruption.

Genetic variability further contributes to differences in caffeine metabolism. Variants in the CYP1A2 and ADORA2A genes influence absorption, metabolism, and physiological responses [41]. Individuals homozygous for the A allele of CYP1A2 produce more cytochrome P450, the enzyme responsible for ~95% of caffeine metabolism, and metabolize caffeine more rapidly [41]. These “fast metabolizers” may experience enhanced ergogenic effects [41,42], although some studies report no genotype-performance relationship [43]. Approximately 46% of the general population are fast metabolizers [41,44], compared with ~59% in athletic cohorts [45]. A recent systematic review and meta-analysis showed that caffeine can have ergogenic effects across a range of performance-related outcomes, including aerobic and muscular endurance, maximal strength, power, and jumping ability [46], but inter-individual variability remains substantial. While most athletes experience performance benefits, some are non-responders, and a minority may even exhibit negative responses [47]. Habitual caffeine use can lead to some tolerance and a blunted ergogenic response, likely due to adaptations in adenosine receptor sensitivity. Nevertheless, most habitual consumers still experience performance benefits, although the magnitude may be reduced. Chronic intake above ~400 mg/day may also increase the risk of adverse effects such as sleep disruption and anxiety, and strategies such as short-term caffeine withdrawal before competition may help restore responsiveness.

In applied settings, practitioners should experiment with different doses, timing strategies, and delivery formats (e.g., anhydrous caffeine, tablets, beverages, gels, or gum). Chewing gum is often preferred by players who experience gastrointestinal discomfort with liquid forms, as buccal absorption allows faster uptake and avoids digestion-related delays [48]. It also mitigates reduced splanchnic blood flow during intense exercise, which can impair absorption from other oral formulations [49]. Caffeine tablets or gum are often used by non-starting players to achieve rapid stimulation before entering a match. Regardless of delivery method, regular monitoring of sleep quality and overall response is recommended.

Most elite players have prior experience with caffeine, but caution is warranted for younger athletes. For players aged 12–17, caffeine intake should generally remain below 100 mg/day, and dosing strategies should be carefully tested and adjusted, a challenge in academy environments with limited nutritional support [49]. Ultimately, individualized experimentation during training, before implementing strategies in competition, allows practitioners and athletes to identify the most effective and tolerable caffeine protocols for optimal performance.

### 3.2. Beta-Alanine

Carnosine is an important intracellular buffer that reduces the accumulation of protons within contracting muscles during high-intensity exercise [50,51]. This buffering action helps delay the onset of fatigue and can improve performance in activities lasting between 1 and 4 min. Direct carnosine supplementation, however, is not an effective strategy to elevate intramuscular carnosine content due to rapid degradation by carnosinase, an enzyme that catalyzes carnosine breakdown in the bloodstream [52]. Instead, carnosine levels can be effectively increased by supplementing with beta-alanine, the rate-limiting precursor in carnosine synthesis.

Research consistently shows that beta-alanine supplementation at doses of 3.2–6.4 g per day, over periods ranging from 4 to 24 weeks, significantly increases muscle carnosine concentrations [53,54,55]. Similarly, supplementation with 65 mg/kg of body mass, typically delivered using a split-dose protocol of 0.8–1.6 g every 3–4 h for 10–12 weeks, has been shown to enhance intramuscular carnosine levels [56].

In a study of 17 highly trained soccer players, Saunders et al. (2012) [57] demonstrated that 12 weeks of beta-alanine supplementation (3.2 g/day provided as four 800 mg sustained-release tablets) significantly improved performance in the Yo-Yo Intermittent Recovery Test Level 2. This improvement was likely attributable to increased muscle buffering capacity, which attenuated the decline in intracellular pH during repeated high-intensity exercise [57]. More recently, Rosas et al. (2017) [58] reported that beta-alanine supplementation (4.8 g/day divided into six equal doses of 0.8 g every two hours for six weeks), combined with plyometric training, further enhanced endurance, repeated sprint ability, and jumping performance in female soccer players.

Although these findings are promising, other studies have reported no significant performance benefits from beta-alanine supplementation [50]. Overall, the current evidence base for beta-alanine use in soccer players remains limited and inconsistent. While a small number of studies have demonstrated improvements in buffering capacity and high-intensity performance, several others have found no significant effects on soccer-specific outcomes. Given the small number of available studies and the variability in results, beta-alanine should be considered an emerging supplement with potential, but it cannot yet be strongly recommended for routine use in soccer without further investigation. Additional research is needed to better understand the mechanisms through which beta-alanine may influence performance, particularly in sport-specific contexts. From a practical perspective, practitioners should also consider potential side effects. Some athletes report immediate tingling sensations (paresthesia) following beta-alanine ingestion, a side effect documented by Chung et al. (2012) [59] in elite swimmers undergoing a 10-week supplementation protocol. This transient tingling can reduce compliance if athletes are not adequately informed. Practitioners should reassure players that the sensation is temporary and harmless and emphasize the potential benefits of beta-alanine as an intramuscular buffer that can delay fatigue. The occurrence of paresthesia is dose-dependent and is associated with both the magnitude and timing of peak blood beta-alanine concentrations [60]. While generally considered benign, strategies such as splitting doses throughout the day, consuming them with meals, or using sustained-release formulations can minimize the severity of this side effect [54].

Despite mixed results, soccer players may still benefit from beta-alanine supplementation as a means of increasing muscle carnosine levels and reducing proton accumulation, one of the primary contributors to fatigue during high-intensity exercise [61]. Such buffering effects are particularly relevant for efforts lasting between 1 and 4 min [62]. As the frequency and intensity of these high-intensity actions have increased in modern soccer [9], beta-alanine supplementation may offer performance advantages in match-deciding activities such as pressing opponents, executing curved sprints, making rapid changes of direction for tactical adjustments, and performing repeated jumping tasks.

### 3.3. Creatine

Creatine is primarily stored in skeletal muscle and the brain. It can be synthesized endogenously in the liver, kidneys, and pancreas, with additional amounts obtained through the diet, mainly from meat and fish [63]. Both short-term, high-dose and long-term, low-dose supplementation protocols have been shown to significantly increase muscle creatine and phosphocreatine (PCr) concentrations [64,65]. Elevated PCr levels may directly enhance high-intensity performance by increasing the capacity for rapid ATP resynthesis during short, intense bouts of exercise and by facilitating glycogen conversion to lactate [66]. Creatine also serves as an intracellular pH buffer, contributing to acid–base regulation and plays a role in oxidative metabolism [67].

Kreider et al. (2017) [65] reported that creatine is widely recommended as an ergogenic aid for power- and strength-based athletes, including those participating in soccer, basketball, and American football. A typical supplementation strategy involves a loading phase of 20 g/day (divided into four 5 g doses) for 5–7 days to rapidly saturate muscle creatine stores, followed by a maintenance dose of 3–5 g/day. When combined with a structured training program, creatine supplementation can promote increases in lean mass, muscular strength, and power [65]. In elite soccer players, short-term supplementation (20 g/day for 5 days) has been shown to improve sprint times (10 m and 30 m) and agility performance [68]. Mujika et al. (2000) [69] similarly observed that creatine supplementation (20 g/day for 6 days) improved repeated sprint performance and maintained countermovement jump capacity in highly trained soccer players. In female players, the same regimen improved maximal sprint performance (6 × 35 m) and jump height [70]. Additionally, a longer-term protocol using a lower dose of magnesium creatine chelate (5 g/day) over a 16-week competitive season resulted in improved repeated anaerobic sprint performance [71].

These findings suggest that creatine supplementation may be used to enhance performance during both training and competitive periods, as well as to prevent or minimize performance declines during congested match schedules [9]. By reducing muscle damage and accelerating the recovery of lost force-production capacity, creatine may help maintain high performance across consecutive matches. Daily doses of 3–5 g can also aid post-exercise recovery and are considered safe and well-tolerated for long-term use in healthy individuals [72]. The loading phase can often be omitted if rapid saturation is not required, with similar results achieved through continued daily intake of 3–5 g depending on body mass [73]. Creatine may also enhance post-exercise glycogen resynthesis [74], and for this reason, supplementation is often recommended after matches to maximize its effects on glycogen replenishment. This strategy is particularly relevant for recovery between matches or training sessions, especially during periods of dense competition, but current evidence does not support a significant acute benefit during match play itself, as creatine’s primary ergogenic effects are linked to chronic supplementation and recovery support rather than in-game performance.

Creatine supplementation in combination with resistance training has been shown to increase maximal power and strength by 5–15%, and total work performed during maximal-effort sets by a similar magnitude [72]. Long-term supplementation improves training quality, leading to greater increases in lean mass, strength, and performance [65]. Increased lean body mass enables soccer players to generate more force in a shorter time, enhancing speed, quickness, and agility, qualities critical for acceleration, deceleration, and overall match performance [75]. Greater muscular strength also improves the ability to execute key skills such as sprinting, jumping, and changing direction, which directly influence scoring opportunities and match outcomes [76]. It is important to note that, similar to caffeine, there is considerable inter-individual variability in response to creatine supplementation. While many athletes experience significant increases in intramuscular phosphocreatine stores and subsequent performance benefits, others may exhibit minimal changes. Responsiveness is influenced by factors such as baseline muscle creatine content, muscle fiber composition (with a greater proportion of type II fibers often associated with a stronger response), habitual dietary intake, and training status [77]. In practice, practitioners can often identify non-responders during the initial loading phase, as these individuals typically do not show the expected increase in body mass associated with intracellular water retention. Because creatine is co-transported with sodium into muscle cells, a lack of weight gain during loading suggests insufficient intramuscular creatine accumulation to meaningfully impact performance.

However, protocols involving a loading phase are often associated with transient increases in body mass, primarily due to water retention. Because creatine supplementation increases intracellular water retention, athletes should ensure adequate daily fluid intake to support proper hydration and minimize the risk of gastrointestinal discomfort, cramping, or heat-related issues during training and competition. As a result, players should be educated about this potential effect, as it may influence psycho-physical status, particularly in weight-sensitive positions. Education should also extend to coaching staff, given their influence on player weight management and performance expectations [77]. Although anecdotal reports of gastrointestinal discomfort (e.g., diarrhea, cramping, and nausea) exist, daily creatine supplementation is considered safe and has even been associated with reduced muscle cramping [63]. 

Creatine monohydrate remains the most extensively studied and effective form of creatine supplementation. Although alternative forms such as creatine salts exhibit improved solubility due to lower pH in solution, none have demonstrated superior efficacy compared to monohydrate [63,77]. Supplementation strategies should be individualized, taking into account player-specific responses and tolerability. Creatine is typically mixed with water, juice, or a recovery beverage or ingested in capsule form with meals. For players who experience gastrointestinal issues, alternative delivery formats, such as incorporating creatine into snacks (e.g., protein bars or chocolate-based recovery products), can improve compliance.

The timing of creatine ingestion appears to have little impact on its overall efficacy, with both pre- and post-exercise supplementation producing similar outcomes [78]. However, emerging evidence suggests that post-exercise ingestion may offer slightly greater benefits [79]. Additional studies are needed to determine the optimal timing of creatine consumption relative to exercise to maximize performance outcomes. Co-ingestion of creatine with carbohydrates or a carbohydrate-protein mixture that stimulates insulin release can further enhance creatine uptake [80].

### 3.4. Sodium Bicarbonate

Sodium bicarbonate supplementation enhances extracellular buffering capacity and may improve performance in activities where acid–base disturbances from anaerobic glycolysis limit output. These include repeated high-intensity sprints and continuous high-intensity efforts lasting between 1 and 7 min [81,82]. Oral ingestion of 200–300 mg/kg body mass of sodium bicarbonate can significantly and safely elevate endogenous bicarbonate levels [83,84]. Although the precise physiological mechanisms underlying its ergogenic effects remain unclear, meta-analyses have reported performance improvements of approximately 2–3% across various metrics, including power output, speed, work capacity, and time to exhaustion, during single and repeated bouts of high-intensity exercise lasting 1–10 min [82,85,86].

A recent consensus statement from the International Olympic Committee (IOC) recommends ingesting 200–400 mg/kg body mass of sodium bicarbonate with a small, carbohydrate-dense meal (~1.5 g/kg body mass of carbohydrate) approximately 120–150 min before exercise [1]. Co-ingesting sodium bicarbonate with a carbohydrate-rich meal not only supports blood alkalosis but may also reduce gastrointestinal distress.

In soccer-specific contexts, Kim (2021) [68] demonstrated that supplementation with 0.3 g/kg/day of sodium bicarbonate, divided into four doses over seven days and co-administered with 20 g of creatine, improved 30 m sprint performance and agility (measured by the arrowhead agility drill) in well-trained soccer players. Similarly, Chycki et al. (2018) [87] reported that daily ingestion of 300 mg/kg sodium bicarbonate combined with 300 mg/kg potassium bicarbonate for nine days enhanced repeated sprint performance (6 × 30 m sprints with 10 s of recovery) compared with placebo.

While such combination strategies may show additive or synergistic benefits, they also make it difficult to isolate the specific contribution of each supplement. Outcomes may vary widely depending on study design, the type of performance task, and individual variability in physiological response [88]. Moreover, distinguishing individual responsiveness from normal day-to-day fluctuations in performance presents an additional challenge.

Although sodium bicarbonate is generally well tolerated at smaller doses, some recreationally active individuals and athletes report gastrointestinal discomfort, including diarrhea, cramping, and bloating [89]. Dividing the total daily dose into smaller portions and ingesting a reduced dose (~4 g) 2–3 h before exercise can help mitigate these side effects. Alternative delivery formats, such as tablets or capsules, may improve palatability and tolerability. Enteric-coated capsules, which resist degradation in the acidic stomach environment, have been shown to further reduce gastrointestinal symptoms [90]. A recent hydrogel-based formulation of sodium bicarbonate, designed to minimize gastrointestinal discomfort and reduce the need for split dosing, has shown promising results. By protecting sodium bicarbonate from stomach acid, it improves gastrointestinal tolerance and ease of ingestion while maintaining its ergogenic potential in high-intensity and repeated-exercise performance [91].

### 3.5. Dietary Nitrate

Dietary nitrate (NO_3_^−^), commonly obtained from vegetables such as spinach, celery, and beetroot or from concentrated beetroot juice, is converted by oral bacteria and enzymatic activity in the digestive system into nitrite (NO_2_^−^) [92]. Nitrite can then be further reduced to nitric oxide (NO), a gaseous signaling molecule involved in several key physiological processes, including vascular tone regulation, neurotransmission, mitochondrial respiration, and skeletal muscle contractile function [93].

Early research indicated that nitrate supplementation could enhance exercise performance, findings supported by many subsequent studies, though not universally [94]. Increased nitric oxide availability is believed to enhance blood flow and nutrient delivery to active muscles, thereby improving exercise tolerance, recovery capacity, and overall performance [95]. In a study involving highly trained male soccer players, supplementation with beetroot juice twice daily (150 mL per serving, totaling ~500 mg of nitrate per day) for seven days, including three days before, on the day of, and three days after a match simulation, attenuated post-exercise decrements in countermovement jump height (at 24 and 48 h), maximal voluntary contraction (at 0, 24, and 48 h), and 20 m sprint performance (at 48 h) following the Loughborough Intermittent Shuttle Test. Compared to the placebo (nitrate-depleted beetroot juice), the decline in performance was significantly reduced [96].

However, evidence is not entirely consistent. A meta-analysis of 80 studies reported no significant ergogenic effect of dietary nitrate in well-trained endurance athletes [95]. Training status appears to be a key factor influencing the efficacy of nitrate supplementation: while well-trained individuals may experience minimal performance benefits [97], studies in untrained or moderately trained subjects have shown improved tolerance to both aerobic and anaerobic exercise. Supporting this, Nyakayiru et al. (2017) [98] demonstrated that six days of beetroot juice supplementation (~800 mg nitrate per day, 140 mL per dose) significantly increased plasma and salivary nitrate and nitrite concentrations. This resulted in a 3.4 ± 1.3% improvement in Yo-Yo Intermittent Recovery Test Level 1 (Yo-Yo IR1) performance (from 1574 ± 47 m to 1623 ± 48 m) and a reduction in mean heart rate compared with placebo [98]. The Yo-Yo IR1 is a standardized field-based assessment widely used in soccer to evaluate an athlete’s ability to repeatedly perform high-intensity exercise interspersed with brief recovery periods. This capacity is crucial in match play, where players must execute repeated sprints, accelerations, and decelerations over extended periods. Performance in the Yo-Yo IR1 is strongly associated with match-related physical demands, making it a relevant indicator of aerobic and anaerobic fitness in soccer players.

Beyond this, several additional studies support a potential role for nitrate supplementation in soccer-specific contexts. For example, supplementation with L-citrulline DL-malate combined with dietary nitrates from amaranth increased maximal sprint speed in female professional players competing in the Spanish first and second divisions. A recent investigation in Slovenian first-division players also showed that nitrate supplementation improved performance when baseline dietary nitrate intake was below ~300 mg/day, though no effects on perceived exertion were observed. Furthermore, survey data from elite soccer nutritionists indicate that nitrate supplementation is recommended by ~50% of practitioners, underscoring its perceived utility despite mixed findings. Together, these studies suggest that nitrate supplementation may confer benefits under certain conditions, particularly when baseline dietary nitrate intake is low or when match demands involve repeated high-intensity actions, but the evidence base remains relatively small and heterogeneous, highlighting the need for further research to optimize dosing strategies, timing, and application in elite soccer.

Dietary nitrate is generally considered safe, with few adverse effects reported. However, some players may experience mild gastrointestinal discomfort [99]. Another common but benign side effect is beeturia, the reddish discoloration of urine following beetroot consumption, which may alarm players unfamiliar with this phenomenon and potentially discourage continued use [100]. Practitioners should educate athletes about beeturia and its harmless nature to improve adherence. Since urine discoloration can interfere with hydration monitoring based on urine color, alternative methods of hydration assessment may be necessary during ni-trate supplementation periods.

### 3.6. Glycerol

Fluid consumption during soccer matches is inherently limited by game structure, and athletes may voluntarily restrict fluid intake due to gastrointestinal discomfort or a diminished thirst drive. These factors increase the risk of hypohydration, a condition characterized by reduced total body water. Scientific debate persists because laboratory studies, which typically involve controlled fluid restriction and standardized exercise protocols, consistently show that hypohydration exceeding ~2% of body mass impairs performance, especially during endurance exercise in hot environments. By contrast, applied studies conducted under real-world competitive conditions sometimes report smaller or inconsistent effects, likely due to behavioral and motivational compensations by athletes or the ability to self-regulate exercise intensity. In prolonged aerobic exercise conducted in temperate or hot environments, performance consistently declines when hypohydration reaches or exceeds ~2% of body mass. However, in cooler conditions or when thirst is effectively mitigated, aerobic performance may be maintained. Reductions in power, strength, and anaerobic endurance are also consistently observed when fluid losses surpass 2% of body mass.

While even a mild level of hypohydration, as little as a 1% loss of body mass, can impair cognitive function, affecting task performance, reaction time, short-term memory, and mood [101], these effects may further compound the physical performance decrements associated with greater fluid losses. One potential strategy to mitigate hypohydration and optimize fluid balance is pre-exercise hyperhydration, a technique that increases total body water before exercise. Glycerol, a water-binding osmotic agent, can enhance hyperhydration efficacy by promoting fluid retention beyond what is achievable with water alone [102]. When combined with sodium, glycerol reduces urine production and increases total fluid retention [103]. This effect is attributed to glycerol’s reabsorption in the renal tubules, which elevates the concentration gradient in the renal medulla and enhances water reabsorption within the nephron [104,105].

Effective glycerol-induced hyperhydration protocols typically involve ingesting 1.2–1.4 g/kg body mass of glycerol dissolved in approximately 25 mL/kg body mass of fluid, consumed 90–180 min prior to exercise [106]. Compared with other hyperhydration approaches, glycerol supplementation is generally well tolerated and associated with minimal gastrointestinal discomfort [107]. Glycerol can also be used post-exercise to support rehydration by adding 1.0 g/kg body mass to every 1.5 L of ingested fluid.

While glycerol-induced hyperhydration has demonstrated benefits such as increased total body water, reduced thermal strain, and modest endurance improvements in controlled exercise models, direct evidence in soccer players remains scarce. Most data come from endurance or intermittent-exercise studies outside of soccer, so any application in this sport should be considered context-dependent (e.g., hot environments, extended matches, or congested competitive schedules) and should be tested in training before competitive use.

Because individual responses to hyperhydration strategies can vary, it is essential for athletes to practice glycerol protocols during training to assess their efficacy and tolerability in real-world soccer scenarios. However, practitioners should also be aware of potential drawbacks. Glycerol hyperhydration typically leads to acute increases in body mass (~0.5–1.0 kg) due to increased fluid retention, which may impair repeated high-speed actions. Some athletes also report headache, nausea, gastrointestinal discomfort, or a bloated sensation. If fluid and sodium intake are not appropriately balanced, there is a theoretical risk of dilutional hyponatremia. Careful monitoring of body mass, symptoms, and electrolyte balance is therefore recommended when implementing glycerol protocols.

It is also worth noting that sodium alone, when consumed with adequate fluid before exercise, is a practical and effective strategy to promote fluid retention and maintain plasma volume. Glycerol may provide an additional osmotic effect that further increases total body water, but the incremental performance benefit of combining sodium and glycerol compared with sodium alone remains uncertain in soccer-specific contexts. For this reason, many practitioners prioritize sodium-based hydration strategies and reserve glycerol use for specific situations, such as competition in hot environments or tournaments with short recovery intervals.

When used according to recommended protocols, for example, an 85% glycerol solution providing 1.2 g/kg body mass, glycerol is considered safe and well tolerated, with a very low incidence of adverse side effects [108]. 

### 3.7. Protein

Dietary protein plays a fundamental role in post-exercise recovery and physiological adaptation, supporting muscle repair, remodeling, and growth. Adequate protein intake can also reduce exercise-induced muscle damage (EIMD) and delayed-onset muscle soreness (DOMS), thereby helping to attenuate performance decrements between matches [109]. However, perspectives on protein’s role in enhancing athletic performance often differ based on the athlete’s training profile. Those engaging predominantly in resistance-based training typically consume higher amounts of protein to support muscle hypertrophy and strength development, whereas endurance athletes may prioritize other nutritional strategies. Nevertheless, there is strong scientific justification for protein intakes that exceed the standard recommended dietary allowance [110,111].

In well-trained soccer players, supplementation with soy or whey protein to achieve a total daily protein intake of approximately 1.5 g/kg of body mass has been shown to improve several field performance metrics, including maximum and average running speed, high-intensity running distance, the frequency of high-speed efforts, and the number of intense accelerations and decelerations during consecutive speed-endurance training sessions [1,2]. The ergogenic potential of protein, particularly milk-derived protein supplements, has also been demonstrated under congested match conditions. In a simulated seven-day fixture period with two matches spaced three days apart, milk protein supplementation may enhance recovery kinetics of soccer-specific performance. Improvements have been reported in 10 m and 30 m sprint times, countermovement jump (CMJ) height, and both concentric and eccentric isokinetic peak torque of the knee extensors and flexors compared with a control group [112].

Although much of the existing literature on protein supplementation focuses on muscle mass accrual or recovery from EIMD, additional research is needed to fully elucidate its role in enhancing soccer-specific performance. Protein supplementation represents a safe and effective ergogenic strategy, particularly when dietary intake does not meet current recommendations. Current consensus guidelines suggest a total daily protein intake of 1.4–2.0 g/kg of body mass, with intakes up to 2.2 g/kg potentially beneficial during periods of intense training or congested competition [111]. Whole food sources such as lean meats, fish, eggs, dairy, and legumes should form the foundation of dietary protein intake. However, supplements can provide a convenient and effective means of meeting protein targets, particularly in scenarios where dietary intake is insufficient or impractical, such as immediately after evening matches or during travel.

Protein supplements such as whey, casein, and soy are well absorbed and highly bioavailable, often matching or even exceeding the digestibility of whole-food proteins due to rapid digestion and efficient amino acid delivery. Among these, whey protein is generally considered the most effective for stimulating muscle protein synthesis because of its high leucine content and rapid absorption kinetics, whereas casein provides a slower, more sustained amino acid release that may be advantageous for overnight recovery. Plant-based proteins, including soy, pea, and rice, can serve as effective alternatives to animal-derived sources. While individual plant proteins often have a slightly lower anabolic potential due to limiting essential amino acids, combining complementary sources (e.g., pea and rice) can result in a complete amino acid profile that closely matches that of animal proteins. Moreover, the inclusion of probiotics and/or digestive enzymes may further enhance digestibility and amino acid absorption, thereby narrowing the gap in quality and functional outcomes compared with animal-based proteins.

Chronic protein intakes substantially exceeding ~2.5–3.0 g/kg/day have not been shown to provide additional performance benefits and may displace other essential macronutrients. While excessive protein intake could potentially exacerbate kidney stress in individuals with pre-existing renal disease, current evidence indicates no adverse effects in healthy athletes.

From a practical standpoint, protein supplementation also offers logistical advantages. For example, following evening matches, players may find it easier and more feasible to consume a protein shake rather than prepare and eat a full meal containing whole-food protein sources such as meat or fish. Such strategies can help ensure the timely delivery of amino acids to muscle tissue, thereby optimizing recovery and adaptation processes.

### 3.8. Tart Cherry

Tart cherry juice has gained attention as a recovery aid for athletes, primarily due to its high concentrations of phenolic compounds and anthocyanins, which possess potent antioxidant and anti-inflammatory properties [113]. These bioactive components can modulate oxidative stress and systemic inflammation, particularly in the period following intense exercise. By enhancing the continuum between exercise and recovery, tart cherry supplementation has the potential to indirectly support physical performance by improving an athlete’s readiness profile and capacity to sustain training and competition demands [1].

Despite growing interest, evidence in soccer players remains limited, with only two published studies to date. Bell et al. (2016) [114] reported that supplementation with Montmorency tart cherry concentrate accelerated recovery following prolonged, repeated-sprint activity in semiprofessional soccer players. In contrast, Abbott et al. (2020) [115] found that tart cherry juice did not significantly enhance recovery following a competitive match in professional players. These conflicting findings may be attributed to differences in participant characteristics (e.g., training status and competitive level), experimental design, or exercise protocols used across studies.

Given the limited and inconsistent evidence base, further research is warranted to clarify whether tart cherry supplementation can reliably improve performance, recovery kinetics, or muscle function in elite soccer populations. Nevertheless, applied insights from practitioners indicate that many players consume tart cherry concentrate following evening matches, a practice supported by emerging evidence suggesting that tart cherry supplementation may improve sleep quality and duration, a critical factor in post-match recovery and adaptation. 

### 3.9. Translating Evidence into Practice: Insights from Practitioner Survey

The practitioners surveyed had an average of 6 ± 4 years of experience working with professional soccer players. Among them, seventy-five percent of practitioners reported working with soccer players on a daily basis, while the remaining 25% indicated working with them less frequently, typically once per week or during scheduled training sessions. Figure 1 shows the distribution of survey participants across the five highest-ranked European soccer leagues, with the largest proportion coming from the Italian Serie A (30%), followed by the English Premier League (25%), Spanish La Liga (20%), German Bundesliga (15%), and French Ligue 1 (10%).

Among them, 55% were nutritionists, 20% were certified strength and conditioning specialists, 10% were physicians, 10% were sport scientists, and 5% served as heads of performance, a role distinct from the coaching staff and typically responsible for coordinating multidisciplinary support to optimize player readiness (Figure 2).

Almost all participants (93.5%) reported an interest in performance supplements, and the majority also reported their use in practice. Regarding perceptions of efficacy and safety, 63.5% of practitioners believed that such supplements are both effective and safe. Widespread recommendations (>80%) were reported for creatine (95%), caffeine (92%), and protein powders (87.5%), whereas dietary nitrate (48.5%), tart cherry (32%), beta-alanine (30%), and sodium bicarbonate (24%) were only moderately recommended. Glycerol (10.5%) was the least frequently recommended supplement (Figure 3).

Protein powders were widely accepted and regularly used by players, with 85% of practitioners recommending protein supplementation post-exercise, often in combination with carbohydrates and creatine, rather than relying solely on whole-food sources. All practitioners (100%) who recommended creatine did so in the monohydrate form. Most (80%) favored a low-dose daily protocol (3–5 g/day), while 20% used a loading strategy (0.3 g/kg/day for 5–7 days) followed by maintenance dosing (0.03 g/kg/day for 4–6 weeks). Creatine was most commonly administered after training (60%), followed by before training (20%) and with meals (20%).

Caffeine supplementation was primarily reserved for match days, typically consumed 30–60 min before kick-off (98%). Reported doses included 100 mg (35%), 200 mg (30%), 150 mg (15%), 250 mg (10%), and >250 mg (10%), provided in the form of gums (50%), pills (25%), or carbohydrate-electrolyte gels (25%). Additionally, 35% of starters consumed 50–80 mg of caffeine at half-time, mostly via energy shots or gums.

Dietary nitrates were predominantly supplemented on match days (65%), with dosages ranging from 400 to 800 mg (73%), most commonly administered in the form of beetroot juice or concentrated nitrate beverages. Beta-alanine was generally supplemented over a 4–8-week period at 3 g/day, typically split between two main meals (85%) and mainly in pill form (90%).

Although the scientific literature identifies several supplements with robust ergogenic or recovery benefits, the survey reveals a notable gap between controlled research findings and applied practice in elite soccer. Supplements with strong and consistent evidence, such as creatine and caffeine, were widely implemented, whereas those with emerging or mixed evidence, such as beta-alanine, tart cherry, and sodium bicarbonate, were only partially integrated into daily routines. These discrepancies reflect the influence of gastrointestinal tolerance, individual preferences, cultural beliefs, and logistical considerations on decision-making in real-world settings. This interpretation is supported by our survey results, in which practitioners frequently cited gastrointestinal discomfort, players’ reluctance to use unfamiliar products, cultural attitudes toward supplementation, and logistical barriers (e.g., dosing complexity and timing) as key reasons for not adopting certain evidence-based interventions. They may also be driven by gaps in knowledge translation, including limited dissemination of scientific findings to practitioners, insufficient familiarity with newer research, and the lack of clear practical guidelines for implementing certain supplements into team-based nutrition strategies.

For example, sodium bicarbonate, despite meta-analytic evidence supporting its benefits for repeated high-intensity exercise, was recommended by only 24% of practitioners. While gastrointestinal side effects and the complexity of timing protocols are frequently cited in the literature as barriers to its use, our findings also highlight a broader challenge, that evidence from research is not always effectively disseminated to or integrated by practitioners, which may contribute to the low adoption of this supplement in applied settings. Similarly, dietary nitrate and tart cherry, which have shown potential to improve exercise tolerance and recovery, were used less frequently, indicating a degree of hesitancy toward interventions perceived as novel or inconsistent. Finally, glycerol, despite its theoretical application as a hyperhydration strategy in hot environments, was seldom recommended (<15%), likely due to limited familiarity in soccer-specific contexts.

These findings suggest that, while scientific evidence provides a foundation for supplement efficacy and its relation to soccer performance, the translation of this evidence into elite soccer settings is often limited, potentially due to gaps in practitioner knowledge, lack of familiarity with certain supplements, or practical barriers to implementation.

Table 1 provides a comparison of scientific evidence and survey-based practitioner use of dietary supplements in elite soccer.

## 4. Discussion

Theoretically, it is straightforward to design evidence-based supplementation strategies based on the available literature. However, significant challenges often arise when attempting to implement these strategies effectively in real-world settings such as the dressing room. Soccer teams are composed of players of diverse nationalities, ages, cultures, religions, and levels of professional experience, each bringing unique daily habits and beliefs that can influence dietary supplement strategies. For example, Muslim players observing Ramadan must abstain from eating and drinking from dawn to dusk, creating additional challenges for supplement planning [116].

Cultural background, nationality, and playing position can all affect dietary habits and adherence to supplementation protocols. Hulton et al. (2022) [117] reported that English players consumed significantly more protein (2.4 g/kg body mass per day) than Dutch or Scottish players, reflecting differences in nutritional beliefs and approaches to muscle recovery. Similarly, Iglesias-Gutiérrez et al. (2012) [118] found notable variations in red meat consumption, with forwards consuming more than players in other positions. Understanding these preferences is essential for performance nutritionists, as it enables the accurate assessment of macronutrient and micronutrient intake and facilitates the development of individualized supplementation plans.

Beyond cultural and personal factors, emotional and physiological barriers also affect supplement adherence. The dynamic and unpredictable nature of soccer, with outcomes ranging from victory to defeat, can influence appetite and willingness to ingest supplements. Matches with high stakes (e.g., relegation battles, championship deciders, European finals, or World Cup fixtures) may heighten stress and focus, causing some players to avoid supplements, while others may increase their intake of caffeinated products to cope with pressure. Gastrointestinal symptoms such as dizziness, nausea, or cramps further complicate supplement use during or after matches [119]. 

To address post-exercise appetite loss, liquid-based nutrient sources, such as milkshakes or meal replacement drinks, can be offered to players unwilling to consume solid food [120]. Similarly, capsules or tablets containing isolated nutrients (e.g., creatine) can provide a more personalized, targeted approach to supplementation. Cultural attitudes toward supplementation represent another important barrier. Many players continue to believe that all necessary nutrients should be obtained exclusively from food and remain skeptical of supplements [121,122,123]. It is crucial to distinguish between supplements designed to meet nutrient requirements and those aimed at enhancing performance. Close et al. (2022) [72] emphasized that, whenever possible, nutrients should be obtained from whole foods and beverages rather than isolated components or supplements. However, practical limitations often make this difficult [124].

For instance, while tea, coffee, and cocoa naturally contain caffeine, determining the exact caffeine content of a beverage is difficult due to variability in brewing time, bean type, roast level, and water-to-coffee ratio [125]. Achieving an ergogenic dose of beta-alanine through diet alone is also impractical: consuming 200 g of chicken and 150 g of turkey yields plasma levels equivalent to just 800 mg of beta-alanine, whereas an effective daily dose is 1.6–6.4 g [126,127]. Likewise, obtaining sufficient dietary nitrates from vegetables would require consuming ~300 g of leafy greens or beetroot before competition, which is often unfeasible and can cause gastrointestinal distress [127]. Concentrated nitrate supplements, with known dosages, are therefore a practical alternative. Similar considerations apply to creatine, as foods rich in creatine (e.g., salmon, pork, and steak) contain ~3–6 g per kg when raw [128]. Reaching the effective dose would require consuming ~1 kg of raw meat four times daily, an impractical and potentially unsafe strategy. Cooking meat typically reduces creatine content by approximately 10–30% due to the conversion of creatine to creatinine during the heating process. This further increases the amount of food required to achieve ergogenic doses, reinforcing why creatine monohydrate supplementation is generally the preferred approach. Creatine monohydrate supplementation remains the most efficient solution [72]. 

Another critical concern is the risk of contamination with World Anti-Doping Agency (WADA)-prohibited substances. Although this risk is higher with non-batch-tested supplements, even food products (e.g., meat contaminated with clenbuterol) may pose a risk [129]. Certification processes reduce, but do not eliminate, this risk, as it is impossible to test for all banned substances [130]. Nevertheless, certified supplements offer a safer alternative, particularly when food safety or intake volume is a concern [72]. 

To overcome many of these challenges, multidisciplinary collaboration and education are key. Performance nutritionists and strength and conditioning coaches should work closely with athletes to develop individualized supplementation strategies. Implementing evidence-based educational programs, which teach safe supplement use and guide decision-making through structured frameworks, can support this process. Figure 4 presents a proposed decision-making model, outlining key objectives, evaluation criteria, and alternative strategies for different scenarios.

Nutritional education can be delivered through multiple approaches, including daily activities, interactive workshops, and facilitated discussions, with program design tailored to the learning preferences of athletes [131]. Players typically prefer short, interactive sessions (≤5 min) over long classroom-based lessons. Moreover, generational differences may influence educational strategies: Millennials, for instance, prefer visual and digital content, while Baby Boomers value repetition and longer-form delivery. Adapting content to these preferences, along with visible reminders in locker rooms and training facilities, can help reinforce positive nutritional behaviors [120]. 

It is important to acknowledge that the present review, and much of the available literature, is limited to studies conducted specifically in soccer players. While a larger body of evidence exists on the ergogenic effects of dietary supplements in other athletic populations, differences in physiological demands, match dynamics, and performance outcomes limit the direct applicability of those findings to soccer. This focus on soccer-specific evidence enhances the ecological validity of the conclusions but may also reduce the breadth of available data. Future research should aim to expand the evidence base with larger, long-term, and sport-specific investigations to better inform practice in elite soccer environments.

## 5. Conclusions

This review highlights both the potential and the limitations of dietary supplements in elite soccer. Supplements with strong and consistent evidence, such as caffeine, creatine, and protein, are widely adopted in practice, whereas those with more variable or emerging evidence, including beta-alanine, sodium bicarbonate, dietary nitrates, and tart cherry, remain underutilized. Bridging the gap between scientific findings and real-world application requires careful consideration of cultural context, gastrointestinal tolerance, logistical constraints, player preferences, and adherence. Safety, quality assurance, and the prevention of inadvertent doping also remain critical factors influencing implementation.

Ultimately, dietary supplements should complement, rather than replace, comprehensive nutritional strategies. Future research should prioritize longitudinal studies on supplement adherence, implementation-focused trials within elite club environments, and the development of soccer-specific dosing protocols. By combining scientific rigor with practical adaptability, practitioners and sports scientists can better support players in sustaining peak performance throughout the demanding competitive season.

## Figures and Tables

**Figure 1 jfmk-10-00408-f001:**
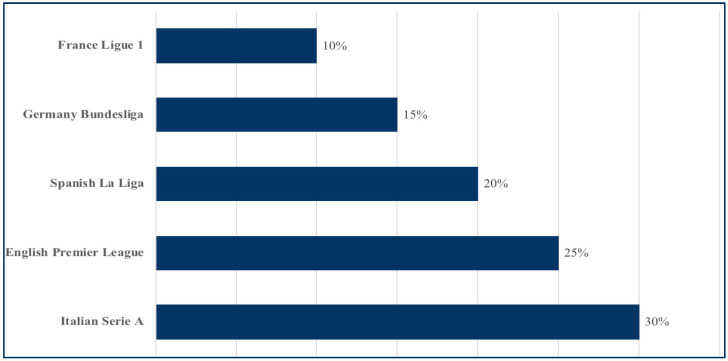
Country of participants.

**Figure 2 jfmk-10-00408-f002:**
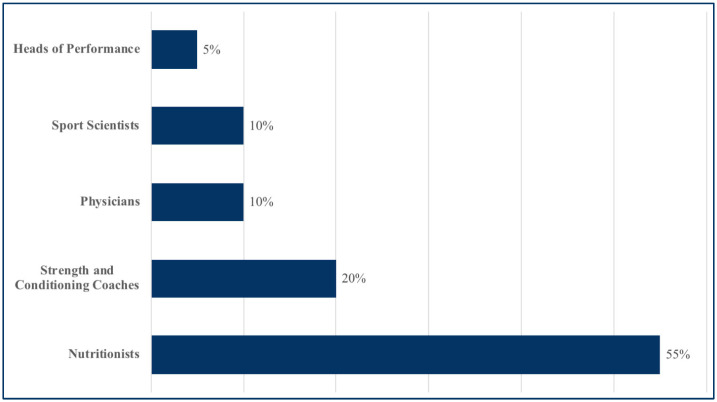
Participants’ job role.

**Figure 3 jfmk-10-00408-f003:**
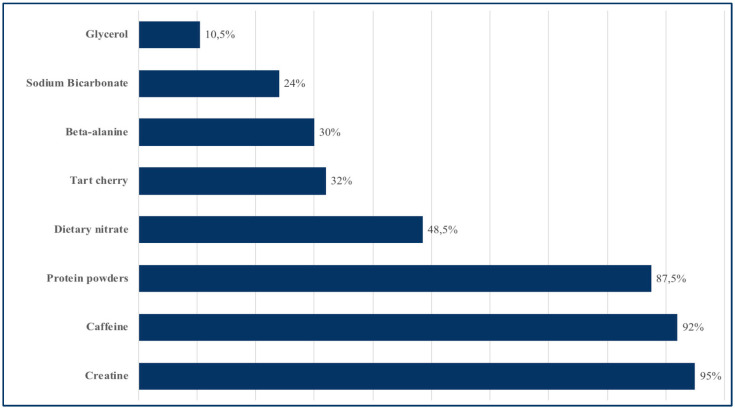
Recommended dietary supplements.

**Figure 4 jfmk-10-00408-f004:**
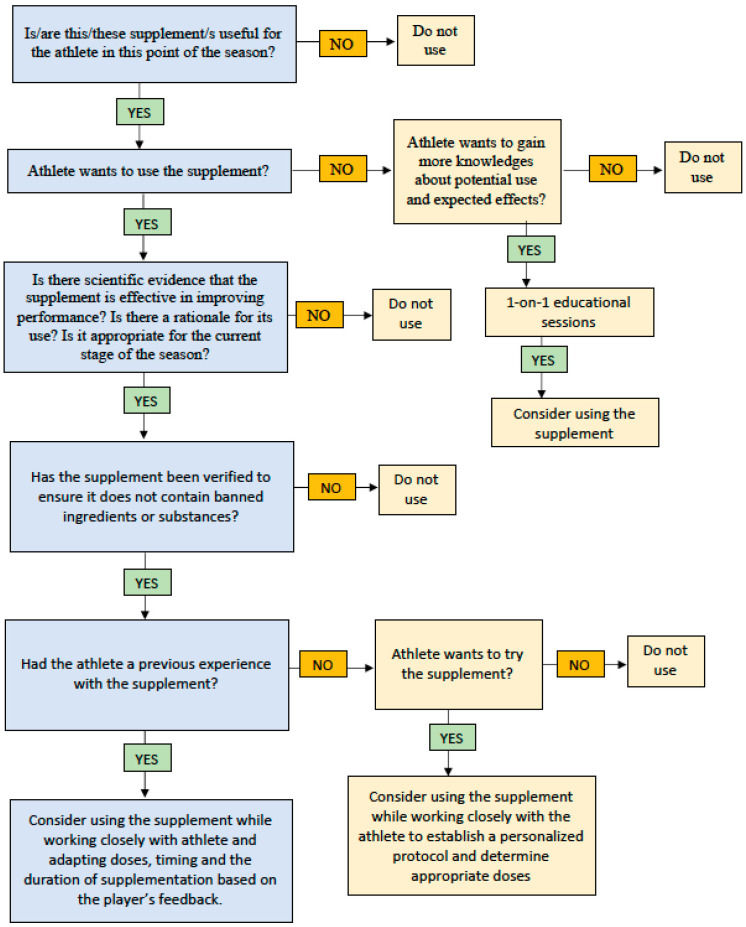
Framework for dietary supplements usage decision-making.

**Table 1 jfmk-10-00408-t001:** Performance supplements and relation to soccer performance.

Supplement	Authors	Evidence from the Literature	Relation to Soccer Performance	Practitioner Use (Survey)	Main Barriers in Practice
Caffeine	Foskett and Gant (2009) [28]	Ingestion of caffeine (6 mg/kg) 60 min before a match improved passing accuracy and jump height.	Enhanced passing accuracy helps midfielders deliver precise passes and crosses, increasing goal-scoring opportunities. Improved jump height enhances central defenders’ aerial ability in duels such as headers.	Widely used (>90%), typically on match days, 30–60 min before kick-off. Doses usually range from 100 to 250 mg, administered as pills, gums, or gels.	Individual variability (fast vs. slow metabolizers), sleep disruption (especially for late matches), and gastrointestinal discomfort from drinks or gels.
Caffeine	Guerra Jr. et al. (2018) [29]	Plyometric and sled-towing stimuli combined with caffeine (5 mg/kg) enhanced jump performance.	Improved jump performance increases aerial effectiveness for defenders and forwards, particularly in headers and duels.		
Caffeine	Nakamura et al. (2020) [30]	Caffeine ingestion (3 mg/kg) improved intermittent sprint ability.	Enhanced sprint ability supports full-backs and wing-backs during high-intensity runs, such as overlapping or making runs behind defenders.		
Caffeine	Ranchordas et al. (2018) [31]	Chewing caffeinated gum (200 mg) for 5 min before performance tests improved countermovement jump, sprint, and Yo-Yo Intermittent Recovery Test Level 1 results.	Improved explosive strength (e.g., countermovement jump) and sprint performance, crucial for key soccer actions both in and out of possession.		
Caffeine	Yildirim et al. (2023) [32]	Chewing caffeinated gum (200 mg) for 10 min improved quadriceps strength, ball-kicking speed, and countermovement jump performance.	Enhanced kicking speed, quadriceps strength, and jump performance, all essential for powerful shots, precise passes, and aerial duels.		
Beta-alanine	Saunders et al. (2012) [57]	Twelve weeks of beta-alanine supplementation (3.2 g/day) improved Yo-Yo Intermittent Recovery Test Level 2 performance.	Enhanced muscle buffering capacity and reduced intracellular pH support high-intensity efforts and improve recovery during games.	Moderately used (~30%), typically short-term (4–8 weeks), taken with meals, often in pill form.	Tingling side effects, variable efficacy, and long supplementation period required.
Beta-alanine	Rosas et al. (2017) [58]	Beta-alanine supplementation (4.8 g/day) combined with six weeks of plyometric training enhanced endurance, repeated sprinting, and jumping ability.	Enhanced repeated sprinting ability can increase the number of attacking sprints, leading to more goal-scoring opportunities.		
Creatine	Muijika et al. (2000) [69]	Creatine supplementation (20 g/day for 6 days) improved repeated sprint performance and maintained countermovement jump height.	Enhanced strength, power, and sprint capacity support critical match actions such as winning aerial duels, tackling, and rapid acceleration.	Widely used (>90%). Primarily creatine monohydrate, usually following a low-dose protocol (3–5 g/day). Typically administered after training or matches.	Weight gain due to water retention, gastrointestinal discomfort in some players, and need for coach education.
Creatine	Zajac et al. (2020) [71]	Magnesium creatine chelate (5 g/day for 16 weeks) improved repeated anaerobic sprint performance.	Helps players outpace opponents during counterattacks, sprint to intercept passes, or accelerate into space to finish plays.		
Creatine	Kim (2021) [68]	Creatine (20 g/day) combined with sodium bicarbonate (0.3 g/kg/day) for seven days improved sprint times (10 and 30 m) and agility performance.	Improves players’ ability to outrun defenders during fast attacks or quickly recover possession.		
Sodium bicarbonate	Chycki et al. (2018) [87]	Co-ingestion of sodium bicarbonate (300 mg/kg/day) and potassium bicarbonate (300 mg/kg/day) for nine days improved repeated sprint performance (6 × 30 m).	Enhanced repeated sprint ability aids in breaking through defensive lines through dribbling, through balls, overlapping runs, and other dynamic actions.	Limited use (~24%). Occasionally used with creatine.	Gastrointestinal discomfort (bloating and diarrhea), impractical timing, and poor adherence.
Nitrate	Daab et al. (2021) [96]	Nitrate ingestion (~500 mg/day for seven days) attenuated performance declines in countermovement jump and 20 m sprint after intermittent exercise.	Maintaining repeated jump and sprint capacity is critical for aerial duels, quick recoveries, and regaining balance during play.	Moderately used (~48%). Mostly consumed on match days as beetroot juice or concentrated shots.	Gastrointestinal discomfort, beeturia, compliance issues.
Nitrate	Nyakayiru et al. (2017) [98]	Nitrate ingestion (~800 mg/day for six days) improved intermittent performance and reduced mean heart rate.	Enhances high-speed running capacity during counterattacks, defensive recoveries, and pressing situations.		
Tart cherry	Abbott et al. (2019) [115]	Tart cherry juice concentrate consumed before and after a 90-min match (12 h and 36 h post-match) reduced declines in countermovement jump reactive strength index (RSI).	Maintaining RSI supports explosive actions such as sprints, changes of direction, and vertical jumps, all decisive for match outcomes.	Limited use (~32%), mainly post-match during congested schedules.	Mixed evidence, novelty, and taste preferences.
Protein	Poulios et al. (2018) [112]	Milk protein supplementation improved maximum speed, sprint times (10 and 30 m), and jump height between matches.	Improved neuromuscular performance enhances fatigue resistance and reduces performance deterioration, preparing players for subsequent matches.	Widely used (~87%). Primarily whey or milk proteins, administered post-training or post-match.	Few barriers; widely accepted as safe and convenient.

## Data Availability

No new data were created or analyzed in this study. Data sharing is not applicable to this article.

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
