# Peer review of "From Science to Dressing Room: Dietary Supplements for Elite Soccer Performance"

_jfmk, 2025, doi:10.3390/jfmk10040408_

Round 1
Reviewer 1 Report
Comments and Suggestions for Authors
General:
-Supplements" and "nutrients" should be differentiated. "Protein" and "carbohydrates" are nutrients, but both can be found within supplements. The authors choose to include protein supplements in their analysis (although it is not clear that they have only considered supplement forms of protein), but they do not consider carbohydrates in any form. Please provide justification.
-"The questionnaire consisted of 20 questions, developed based on the domain expertise of the authors." --> Justification should be provided for the questions in the survey. Was there any consideration for a validated survey? I recommend including the survey as an appendix the manuscript. Additionally, the data presented make me wonder what the 20 questions were about.
-The review of literature does not seem comprehensive. I recommend major revisions to this section that consider all available literature for soccer players and supplementation.
-The paper is not organized in the way it should be. Many of the results are included in the Methods section. This needs to be re-organized.
Line 15: Change "is" to "are".
Line 18: Choose a different word other than "enrolling" for "Literature".
Line 20: Change "in" to "with".
"and findings were compared with insights obtained by an online questionnaire addressed to practitioners working in elite soccer teams from five European Leagues." --> Why? This is not standard for a review.
Line 25: What is meant by "congested match conditions"?
Line 27: Add "the" before field.
Lines 28-29: Re-phrase sentence.
Line 29: Change to "Sodium bicarbonate and glycerol were..."
"These findings reveal that while scientific evidence provides a foundation for supplement efficacy, translation into elite soccer requires adaptation to the complexities of the dressing room, taking into account factors such as gastrointestinal tolerance, habits, cultural beliefs, lack of familiarity and individual preferences of players." --> The findings of your study do not reveal the motivations of the practitioners for not implementing the supplements you found were validated in soccer players into practice (at least not based on the provided information). I suspect many of them are unfamiliar with the research about supplements. This aspect needs to be addressed.
Line 39: Do you mean "context"?
Lines 40-42: IOC definition needs to be in quotes
At this point, I have stopped making comments on the English language, which needs to be significantly addressed by the authors throughout the manuscript before this manuscript is seriously considered.
"In elite soccer, dietary supplements are integral to nutritional strategies that support players in enhancing performance and recovery." --> What is mean by this sentence? Are nutritional supplements integral to elite soccer. If that is what you are claiming, I would disagree. Are you saying that they are heavily used in soccer? Please clarify what is meant by this statement.
"It is well documented that the distance covered at higher intensities during matches is a valid measure of physical performance in soccer." --> What do you mean by this?
"Coaches, sports nutritionists, sports scientists and physicians, continuously work to understand the relationship between dietary supplements, nutrient timing, and athletic performance (Hostrup and Bangsbo, 2023). N." --> Is this true? I think many are not well-informed.
Lines 74-79: Is this properly quoted?
"A key aspect of the success of any nutritional intervention is understanding whether a supplement can impact sport-specific performance." --> I would argue that this is not true. Like you just mentioned with carbohydrates, many nutritional strategies do not entail supplements at all.
Line 88: Sports drinks and sports bars are not considered supplements. Please revise.
Lines 115-122: It needs to be clarified somewhere that this definition for professional soccer players does not apply to the literature search.
In the search strategy, why weren't individual supplements of interest used as search terms?
2.2-2.7: Why are these included under the "Materials and Methods" section? These appear to be results.
Caffeine: Are doses above 6 mg/kg deleterious in nature to the performance of soccer players?
Lines 126-129: This sentence is incomplete. Is part of it missing?
Lines 158-159: Enhanced passing accuracy and jumping ability would do more than this, so why limit the context to just one thing for each?
Lines 162-186: Does the research indicate that this would translate to an in-game scenario or just simulated practice?
Lines 179-182: What else was found in the energy drink? Do we know that the results are attributable to the caffeine specifically?
Caffeine: Do metabolism and breakdown depend on the total dose given and the way caffeine was administered (ex: anhydrous solution vs. coffee)?
Caffeine: What about the effects of caffeine from individuals who use it very often? Is their response blunted? Do they have more negative effects? Do they become immune to it? If so, how much is too much?
Lines 194-197: Separate what you mean here by individuals who do not normally consume caffeine and those who consume high doses.
Line 226: What is meant by "performance" here? Aren't these all aspects of performance?
Line 231: Change the word "formats".
Line 242: By "substitute players", do you mean non-starters? Haven't you already addressed caffeinate gum above?
"Yo-Yo Intermittent Recovery Test Level" --> Explain more about what this is and how it applies to soccer players.
Beta-alanine: Are there only 2 studies to support its use in soccer players? How many studies, as referred to in lines 276-277, report no effects in soccer players? Perhaps beta-alanine should not yet be recommended for soccer players if there is not more evidence to support its use.
Lines 338-340: Is there research to support this notion? Could it help recovery more during the game?
Creatine: Discuss the need to increase water consumption so that athletes do not see negative consequences.
Creatine: As with caffeine, the research indicates that there are responders and non-responders to creatine. This should be discussed in this section.
There are 2 sections labelled as "2.7".
Lines 416-417: Specify what is meant by "players" here. Is this specific to soccer players? Is this in athletes in general?
Lines 424-427: Is it effective for sports performance? Is it well-tolerated?
Sodium bicarbonate: Is there only one study that looks at the effects of sodium bicarbonate alone in soccer players?
Lines 434-436: Specify if this is in the context of soccer alone or if it is in all athletic context.
Nitrate: Is there only one study to look at the effects of nitrate in soccer players?
Line 455: "Yo-Yo IR1" --> More needs to be explained about this test.
Line 459: Do you mean "soccer players" here?
"Scientific debate persists regarding the seemingly conflicting results between field studies and laboratory research on the impact of hypohydration on exercise performance." --> Explain what you mean here.
Lines 477-479: This sentence is incomplete. Was the full thought included?
Glycerol: Are there any studies in soccer players at all to justify its use in the sport? Are there potential negative effects of hyperhydration or the weight gain associated with it? You mention the addition of sodium; could the same effects be accomplished with sodium ingestion alone (instead of in combination with glycerol)?
Lines 516-521: Was this through supplementation or milk consumption?
Protein: Are there recommended ranges of consumption? How much should come through supplements vs. normal foods? Are supplements as well absorbed in the body as whole foods? Are there negative consequences of consuming too much protein? Are certain protein supplements more effective?
Lines 237-239: They suggested it? Or, their research supported it?
Lines 555-556: "first men team elite of the first league" --> What does this mean?
"75% worked with soccer players on a daily basis, while 25% engaged with 580 players once a week." --> Daily vs. once per week? It seems unusual that there was nothing in between.
Lines 583-584: Is this the same or different from a coach?
Lines 582-584: This information is redundant with what is presented in Figure 2.
"Almost all participants reported an interest in and use of performance supplements." --> Separate this out because these are two separate questions.
"In contrast, glycerol (10.5%) was not commonly recommended (<20%)." --> Why are there 2 statistics for this line?
Lines 589-596: This is redundant with what is displayed in Figure 3.
"The majority of practitioners recommend protein after exercise (85%) in combination with carbohydrates and creatine." --> Clarify if they recommended protein generally, such as what is found in whole foods, or if they recommended protein supplements specifically.
Lines 601-602: Provided by who?
Do you know that players were only consuming that which was provided to them by practitioners?
How were nitrates administered?
Lines 616-623: The supplementation usage seems to corroborate the evidence. Explain the discrepancies that you allude to.
"These differences underscore the influence of factors such as gastrointestinal tolerance, individual preferences of players and cultural beliefs." --> How do your findings underscore the influence of GI issues, etc.?
"By contrast, sodium bicarbonate, despite meta-analyses showing benefits for repeated high-intensity performance, was recommended by only 24% of practitioners, primarily due to gastrointestinal side effects and challenges with timing protocols." --> Please specific if practitioners indicated that this was the reason for low recommendations. One very important point that seems to be neglected is the common problem that happens where research is conducted but not disseminated to those who might be most likely to use it. This needs to be discussed and considered in the current findings.
"These findings reveal that while scientific evidence provides a foundation for supplement efficacy and relation with soccer performance, translation into elite soccer requires adaptation to the complexities of the dressing room." --> Do the findings show this? Or, do they simply suggest that practitioners are not informed about all of these supplements?
Line 694: Clarify what is meant by a "standardized dose" here. Is this supplemented? If so, what is the formulation?
Line 695: What is mean by "efficient dose" here? Do you mean "effective"?
Lines 708-712: How does this compare to their cooked counterparts?
Line 714: Define WADA
Comments on the Quality of English Language
The English Language needs to be addressed in this manuscript before acceptance.
Author Response
Comments and Suggestions for Authors
Forward
Dear reviewer, we thank you for the important comments. We would like to inform you that all changes have been highlighted in green.
Grammatical errors have been corrected, and the language has been smoothed for clarity, accuracy, and readability. Please note that the changes were not tracked using Track Changes because the extent of the revisions was substantial, and doing so would have made the revised manuscript difficult to read.
General:
#1 Supplements" and "nutrients" should be differentiated. "Protein" and "carbohydrates" are nutrients, but both can be found within supplements. The authors choose to include protein supplements in their analysis (although it is not clear that they have only considered supplement forms of protein), but they do not consider carbohydrates in any form. Please provide justification.
Response: We thank the reviewer for this important observation and agree that a clear distinction between “nutrients” and “dietary supplements” is necessary. In this review, our focus was specifically on dietary supplements as defined by the International Olympic Committee (IOC), namely, “a food, food component, nutrient, or non-food compound intentionally consumed in addition to the habitual diet to confer a specific health or performance benefit.” Based on this definition, we included protein powders because they are widely marketed, consumed, and regulated as dietary supplements, typically provided in concentrated forms such as powders or ready-to-mix formulations that are clearly distinct from whole foods. In contrast, carbohydrates, while undeniably important for performance, are most commonly consumed in the form of foods or beverages (e.g., meals, sports drinks, gels) rather than as standalone dietary supplements (e.g., capsules, tablets, or concentrated powders). As such, they fall outside the defined scope and objectives of this review, which aimed to focus on supplemental products rather than macronutrient-based fueling strategies. We have clarified this rationale in the manuscript to make our inclusion criteria more explicit.
#2 "The questionnaire consisted of 20 questions, developed based on the domain expertise of the authors." --> Justification should be provided for the questions in the survey. Was there any consideration for a validated survey? I recommend including the survey as an appendix the manuscript. Additionally, the data presented make me wonder what the 20 questions were about.
Response: We thank the reviewer for this valuable suggestion. In response, we have substantially revised the Methods section to provide a clearer justification of the questionnaire design and greater detail about its structure and content. The survey was developed based on the subject-matter expertise of the authors and refined in consultation with academic peers and practitioners working in elite soccer to ensure relevance, content validity, and practical applicability. While no existing validated instrument specifically addressed dietary supplement use in elite soccer, our approach followed standard practices for developing targeted questionnaires in applied sport science research. To enhance transparency and reproducibility, we have now included a comprehensive description of the questionnaire’s structure in the Methods section (see Section 2.2). The 20 questions were organized into three domains: (1) participant background information (e.g., age, role, league, experience), (2) perceptions and attitudes toward dietary supplements (e.g., knowledge, interest, demand, safety, and efficacy), and (3) real-world practices (e.g., types of supplements used or recommended, dosing protocols, timing strategies). Examples of individual questions are now provided in the text to give readers a clear understanding of the survey content. We integrated this detailed description directly into the main text rather than as a separate appendix, ensuring that readers can fully evaluate the instrument without the need for supplementary materials.
#3 The review of literature does not seem comprehensive. I recommend major revisions to this section that consider all available literature for soccer players and supplementation.
Response: We appreciate this comment. Our review was intentionally scoped to soccer-specific evidence to maximize ecological validity and practical relevance for elite clubs. As such, we excluded studies conducted exclusively in other athlete populations unless soccer data were reported separately. We recognize that a much larger body of literature exists on these nutrients in non-soccer sports; however, performance demands, positional requirements, and match constraints in soccer can limit the generalizability of findings from other sports. To clarify this scope and strengthen transparency, we have expanded the Methods to state explicitly that only studies conducted in soccer players were included, with clear eligibility/exclusion criteria and search details (databases, terms, and timeframe) and added a brief context paragraph acknowledging the broader sport literature (e.g., consensus statements) without incorporating non-soccer trials as primary evidence. Added a Limitations paragraph highlighting the trade-off between sport-specific focus and breadth. If there are specific soccer-focused studies the reviewer believes we have missed, we would be grateful for the citations and will gladly evaluate and incorporate them where appropriate.
#4 The paper is not organized in the way it should be. Many of the results are included in the Methods section. This needs to be re-organized.
Response: We thank the reviewer for this helpful observation. In the revised manuscript, the section describing the survey design, participant characteristics, recruitment process, and data collection procedures, which was previously misplaced in the Results section, has now been relocated to the Methods section.
#5 Line 15: Change "is" to "are"; Line 18: Choose a different word other than "enrolling" for "Literature"; Line 20: Change "in" to "with"; Line 27: Add "the" before field; Lines 28-29: Re-phrase sentence; Line 29: Change to "Sodium bicarbonate and glycerol were..."
Response: Thank you for these detailed language suggestions. The manuscript has been thoroughly reviewed and revised by a native English speaker. All grammatical errors have been corrected, and the language has been smoothed for clarity, accuracy, and readability. Please note that the changes were not tracked using Track Changes because the extent of the revisions was substantial, and doing so would have made the revised manuscript difficult to read. We have provided detailed explanations of these changes in our responses to the reviewers to each individual concern.
#6 "and findings were compared with insights obtained by an online questionnaire addressed to practitioners working in elite soccer teams from five European Leagues." --> Why? This is not standard for a review.
Response: We appreciate the reviewer’s comment and understand the concern regarding the inclusion of a questionnaire within a review manuscript. However, this article was intentionally designed as a two-part study to provide a more comprehensive and practically relevant perspective on the topic. First, we conducted a narrative literature review focusing exclusively on clinical studies investigating the effects of dietary supplements in soccer players. This forms the scientific foundation of the manuscript and summarizes current evidence regarding supplement efficacy, safety, and application in this population. Second, we conducted a cross-sectional survey of practitioners working with elite soccer teams across five European leagues to capture real-world practices, beliefs, and implementation strategies regarding supplementation. This complementary component was included to address a well-recognized gap in the field: while many reviews summarize the available evidence, few attempt to understand how, or whether, that evidence is translated into daily practice in professional soccer environments. The Discussion section of the manuscript then integrates these two components by comparing scientific evidence with actual practitioner practices, highlighting areas of alignment, discrepancies, and opportunities for improved knowledge translation. We believe that this combined approach significantly enhances the practical relevance and translational value of the review, providing insights that are useful not only for researchers but also for practitioners, performance staff, and policy makers in elite sport.
#7 Line 25: What is meant by "congested match conditions"?
Response: Thank you for pointing this out. By “congested match conditions” we refer to competitive schedules where players have limited recovery time between matches, for example, playing two matches within a single week during regular league play, or competing in both a domestic league and an international tournament (e.g., Europa League or Champions League) within the same week. We have clarified this in the revised manuscript. “…while glycerol contributes to fluid retention in hot environments and during compressed match schedules, where players compete in multiple matches within a short recovery window.”
#8 "These findings reveal that while scientific evidence provides a foundation for supplement efficacy, translation into elite soccer requires adaptation to the complexities of the dressing room, taking into account factors such as gastrointestinal tolerance, habits, cultural beliefs, lack of familiarity and individual preferences of players." --> The findings of your study do not reveal the motivations of the practitioners for not implementing the supplements you found were validated in soccer players into practice (at least not based on the provided information). I suspect many of them are unfamiliar with the research about supplements. This aspect needs to be addressed.
Response: Thank you for this insightful comment. We agree that the original statement may have overstated the conclusions of our study. Our survey did not directly assess the underlying motivations or reasons why practitioners do not implement certain evidence-based supplements. While we inferred potential barriers (e.g., gastrointestinal tolerance, cultural beliefs, taste preferences) from open-ended responses and existing literature, it is also likely that unfamiliarity with current research, limited exposure to emerging evidence, and lack of formal nutrition education contribute significantly to the gap between science and practice. We have now clarified this limitation in the manuscript and explicitly stated that our conclusions are based on practitioner-reported barriers and plausible explanations supported by previous studies, rather than direct measurement of motivational factors.
#9 Line 39: Do you mean "context"?
Response: Thank you for the helpful suggestion. We have revised the sentence for clarity. It now reads: “Dietary supplements encompass a diverse range of substances, and their definition often varies depending on the regulatory, clinical, or performance context.”
#10 Lines 40-42: IOC definition needs to be in quotes
Response: Thank you for your comment. We have added quotation marks around the IOC definition as suggested.
#11 "In elite soccer, dietary supplements are integral to nutritional strategies that support players in enhancing performance and recovery." --> What is mean by this sentence? Are nutritional supplements integral to elite soccer. If that is what you are claiming, I would disagree. Are you saying that they are heavily used in soccer? Please clarify what is meant by this statement.
Response: Thank you for this valuable comment. Our intention was not to suggest that dietary supplements are essential or indispensable to elite soccer, but rather that they are commonly used as part of broader nutritional strategies aimed at supporting performance and recovery. We have revised the sentence to clarify this point. The revised sentence now reads: “In elite soccer, dietary supplements are commonly incorporated into nutritional strategies designed to support players’ performance and recovery.”
#12 "It is well documented that the distance covered at higher intensities during matches is a valid measure of physical performance in soccer." --> What do you mean by this?
Response: Thank you for your comment. Our intention was to emphasize that high-intensity running distance, typically defined as running above a certain speed threshold, is widely recognized in the literature as a reliable indicator of a player’s physical performance capacity during matches. This metric reflects a player’s ability to sustain repeated bouts of intense activity, which is closely linked to match outcomes and overall physical conditioning. The revised sentence now reads: “High-intensity running distance during matches is a well-established indicator of performance”
#13 "Coaches, sports nutritionists, sports scientists and physicians, continuously work to understand the relationship between dietary supplements, nutrient timing, and athletic performance (Hostrup and Bangsbo, 2023). N." --> Is this true? I think many are not well-informed.
Response: Thank you for this helpful comment. We agree that the original sentence may have overstated the level of knowledge and engagement across all practitioner groups. While some professionals actively investigate and apply evidence-based approaches, others may have limited awareness of the latest research or lack formal training in supplement science. We have revised the sentence to better reflect this reality. “Coaches, sports nutritionists, sports scientists, and physicians play an important role in understanding and applying knowledge about dietary supplements, nutrient timing, and athletic performance, although levels of expertise and familiarity with current evidence can vary considerably across these groups.”
#14 Lines 74-79: Is this properly quoted?
Response: Thank you for this comment. Upon review, we realized that the sentence attributed to Arsène Wenger was a paraphrase rather than a verbatim quotation. To accurately reflect the source and maintain proper academic style, we have removed the quotation marks and rephrased the sentence as indirect speech. It now reads: “For example, Arsène Wenger emphasized that nutrition should support players’ health and overall wellness, help optimize performance by ensuring they are in prime physical condition for each match with optimal body composition, facilitate rapid recovery after matches or intense training sessions, and provide adequate fuel to sustain high-intensity effort for 90 minutes or more (Wenger, 2021).
#15 "A key aspect of the success of any nutritional intervention is understanding whether a supplement can impact sport-specific performance." --> I would argue that this is not true. Like you just mentioned with carbohydrates, many nutritional strategies do not entail supplements at all.
Response: Thank you for this helpful comment. We agree that the original sentence was too narrow and implied that the success of nutritional interventions depends solely on supplementation, which is not the case. Many effective nutritional strategies, such as carbohydrate periodization, protein timing, or overall dietary composition, do not involve supplements at all. To address this, we removed the original sentence and rephrased the idea more accurately. The revised version now reads: “A key aspect of nutritional strategy is understanding how different approaches, including, but not limited to, supplementation, can influence sport-specific performance.”
#16 Line 88: Sports drinks and sports bars are not considered supplements. Please revise.
Response: Thank you for this comment. We agree that sports drinks and sports bars are generally classified as functional foods rather than dietary supplements. To address this, we have revised the sentence to focus only on products typically recognized as supplements. It now reads: “Surveys also indicate that products such as whey protein, caffeine, and creatine remain among the most frequently consumed dietary supplements by elite players”
#17 Lines 115-122: It needs to be clarified somewhere that this definition for professional soccer players does not apply to the literature search.
Response: Thank you for pointing this out. We agree that it is important to clarify that the classification of “professional soccer players” provided in the introduction is used only for contextual purposes and does not represent a criterion applied in the literature search. We have now added a clarifying sentence to address this. We have added the following sentence: “It should be noted that this definition of professional soccer players is provided for contextual understanding of the population discussed and was not used as an inclusion or exclusion criterion in the literature search.”
#18 In the search strategy, why weren't individual supplements of interest used as search terms?
Response: Thank you for this comment. Our primary objective was to capture the full scope of the literature investigating dietary supplements specifically in soccer players, rather than limiting the search to predefined ingredients. Using broad search terms (e.g., “dietary supplement*,” “ergogenic aid*,” “soccer,” “football,” etc.) ensured that we did not unintentionally exclude relevant studies, including those examining less commonly studied or emerging supplements. We then conducted a secondary screening of the retrieved studies to identify and categorize individual supplements investigated within this body of literature. This two-step approach allowed us to provide a more comprehensive overview of supplementation practices and evidence in soccer without introducing selection bias at the search stage.
#19 2.2-2.7: Why are these included under the "Materials and Methods" section? These appear to be results.
Response: Thank you for this helpful observation. We agree that sections 2.2–2.7 were more appropriately classified as results rather than materials and methods. In response, we have revised the manuscript structure accordingly by moving these sections from the “Materials and Methods” to the “Results” section to improve clarity and alignment with standard manuscript organization.
#20 Caffeine: Are doses above 6 mg/kg deleterious in nature to the performance of soccer players?
Response: Thank you for this important question. Most studies investigating caffeine supplementation in soccer and other team sports have used doses in the range of 3–6 mg/kg body mass, which are generally effective for enhancing performance without significant adverse effects. Evidence on higher doses (>6 mg/kg) is limited, but some studies and reviews suggest that they do not confer additional ergogenic benefits and may increase the risk of side effects such as gastrointestinal discomfort, jitteriness, anxiety, or sleep disturbances, all of which can negatively impact performance, especially in repeated-sprint or skill-dependent sports like soccer. To address this, we have added a clarifying sentence to the manuscript: “Importantly, doses exceeding ~6 mg/kg have not consistently demonstrated additional ergogenic effects and may increase the risk of adverse outcomes, such as gastrointestinal distress, anxiety, or impaired sleep, potentially negating any performance benefits.”
#21 Lines 126-129: This sentence is incomplete. Is part of it missing?
Response: Thank you for this comment. We agree that the original sentence was incomplete and unclear. We have revised it to form a complete sentence and clearly describe the search approach. It now reads: “Keywords such as ‘performance supplements’ and ‘soccer’ were used, individually combined with terms including ‘supplementation,’ ‘athletes,’ ‘ergogenic,’ ‘peak performance,’ ‘nutritional strategies,’ and ‘ergogenic aids.’”
#22 Lines 158-159: Enhanced passing accuracy and jumping ability would do more than this, so why limit the context to just one thing for each?
Response: We agree that the original sentence was too narrow and did not fully capture the broader tactical and performance implications of improved passing accuracy and jumping ability. We have revised the paragraph discussing the findings of Foskett et al. (2009) to reflect a more comprehensive understanding of their impact on match performance. The revised text now reads: “Foskett et al. (2009) reported that acute caffeine ingestion (6 mg/kg), consumed 60 minutes before a 90-minute soccer-specific simulation, significantly improved passing accuracy and jump height. Enhanced passing precision can improve overall team coordination, facilitate successful attacking transitions, and increase the likelihood of creating goal-scoring opportunities. Likewise, improved jumping performance not only strengthens a central defender’s aerial presence but also benefits forwards and midfielders during set pieces, contested balls, and transitional phases of play.”
#23 Lines 162-186: Does the research indicate that this would translate to an in-game scenario or just simulated practice?
Response: Most of the studies cited in this section, including those by Guerra Jr. et al. (2018), Nakamura et al. (2020), Ranchordas et al. (2018), and Yildirim et al. (2023), were conducted in controlled or simulated practice settings (e.g., laboratory protocols, soccer-specific simulations, or standardized performance tests) rather than during competitive match play. While these protocols are designed to closely replicate the physical and physiological demands of soccer, they do not fully account for the complex tactical, psychological, and environmental factors present in real match conditions. To address this, we have added a clarifying sentence to the manuscript: “It is important to note that most of the evidence supporting caffeine’s ergogenic effects in soccer comes from controlled or simulated practice settings designed to mimic match demands. Although these findings are highly relevant, further research conducted during actual competitive match play is needed to confirm their direct translation to in-game performance.”
#24 Lines 179-182: What else was found in the energy drink? Do we know that the results are attributable to the caffeine specifically?
Response: You are correct that many of the studies included in the systematic review by Abreu et al. (2023) examined the effects of commercial energy drinks, which often contain additional bioactive ingredients beyond caffeine and carbohydrates, such as taurine, B-vitamins, glucuronolactone, and herbal extracts. Because these ingredients were not always tested in isolation, it is not possible to definitively attribute the observed performance benefits solely to caffeine. However, given that the doses of caffeine used were within the range known to elicit ergogenic effects (typically 3–6 mg/kg), it is likely that caffeine was a primary contributor, even if synergistic or additive effects from other components cannot be excluded. We have added the following sentence to the revised manuscript: “It should be noted that many of the studies included in the review used commercial energy drinks that contained additional ingredients such as taurine, B-vitamins, or herbal extracts. As a result, the observed effects cannot be attributed exclusively to caffeine, although the caffeine content likely played a major role in the ergogenic response.”
#25 Caffeine: Do metabolism and breakdown depend on the total dose given and the way caffeine was administered (ex: anhydrous solution vs. coffee)?
Response: Yes, both the dose and the form of administration can influence caffeine’s absorption kinetics, metabolism, and physiological effects. Higher doses are associated with prolonged half-life and increased inter-individual variability in plasma concentrations. Moreover, the delivery matrix can significantly affect the rate and extent of absorption: for example, anhydrous caffeine (e.g., capsules or tablets) is typically absorbed more rapidly and produces higher peak plasma concentrations than caffeine delivered in coffee or energy drinks, where co-ingested compounds (such as chlorogenic acids or other bioactive substances) may slow absorption or alter metabolism. To address this, we have added the following clarifying sentence to the manuscript: “Caffeine metabolism and absorption kinetics can be influenced by both the total dose ingested and the form of administration; higher doses tend to prolong half-life, while anhydrous preparations generally produce faster absorption and higher peak plasma concentrations than coffee or energy drinks, in which other bioactive compounds can modulate these effects.”
#26 Caffeine: What about the effects of caffeine from individuals who use it very often? Is their response blunted? Do they have more negative effects? Do they become immune to it? If so, how much is too much?
Response: Thank you for raising this important point. Habitual caffeine consumption can indeed influence the ergogenic response. Evidence suggests that individuals who consume caffeine regularly may experience some degree of tolerance, leading to a blunted performance response compared with non-habitual users. This is likely due to adaptive changes in adenosine receptor density and sensitivity. However, even in habitual consumers, many studies still report meaningful performance benefits, although the magnitude may be smaller. Chronic high intake — typically defined as >400 mg per day for adults — can also increase the likelihood of negative effects, including sleep disruption, anxiety, gastrointestinal discomfort, and increased heart rate. Nevertheless, “immunity” to caffeine’s ergogenic effects is rarely complete, and tapering or temporarily reducing intake in the days leading up to competition may help restore sensitivity in some individuals. To address this in the manuscript, we propose adding the following sentence: “Habitual caffeine use can lead to some tolerance and a blunted ergogenic response, likely due to adaptations in adenosine receptor sensitivity. Nevertheless, most habitual consumers still experience performance benefits, although the magnitude may be reduced. Chronic intake above ~400 mg/day may also increase the risk of adverse effects such as sleep disruption and anxiety, and strategies such as short-term caffeine withdrawal before competition may help restore responsiveness.”
#27 Lines 194-197: Separate what you mean here by individuals who do not normally consume caffeine and those who consume high doses.
Response: We agree that the original sentence did not clearly distinguish between individuals who are caffeine-naïve or sensitive and those who consume high doses from multiple sources. We have revised the text to make this distinction clearer. It now reads:
“Additionally, individuals who are not accustomed to caffeine use or who are particularly sensitive to its effects may experience adverse reactions such as nervousness, restlessness, headaches, gastrointestinal discomfort, or insomnia even at moderate doses. Those who consume high doses from multiple sources (e.g., coffee, anhydrous caffeine, energy drinks, gels, or gum) are at even greater risk of these side effects, which can include more pronounced sleep disruption, anxiety, and digestive issues (Nédélec et al., 2015).”
#28 Line 226: What is meant by "performance" here? Aren't these all aspects of performance?
Response: Thank you for pointing this out. We agree that the original wording was imprecise, as the listed outcomes, aerobic endurance, muscular endurance, maximal strength, power, and jumping ability, are all aspects of performance. To improve clarity, we have revised the sentence to reflect this: “A recent systematic review and meta-analysis showed that caffeine can have ergogenic effects across a range of performance-related outcomes, including aerobic and muscular endurance, maximal strength, power, and jumping ability (Grgic et al., 2018).”
#29 Line 231: Change the word "formats".
Response: Thank you for this suggestion. We agree that the term “formats” was imprecise in this context. We have revised format to “delivery format”.
#30 Line 242: By "substitute players", do you mean non-starters? Haven't you already addressed caffeinate gum above?
Response: Thank you for your comment. We agree that “substitute players” was unclear. We have revised the sentence to clarify that we are referring to non-starting players who may enter the match later and use caffeine for rapid stimulation. Additionally, we streamlined the text to avoid repetition about caffeinated gum, which had already been discussed earlier. The revised sentence now reads:
“Caffeine tablets or gum are often used by non-starting players to achieve rapid stimulation before entering a match.”
#31 "Yo-Yo Intermittent Recovery Test Level" --> Explain more about what this is and how it applies to soccer players.
Response: Thank you for this valuable suggestion. We have expanded the description of the Yo-Yo Intermittent Recovery Test Level 1 to provide more context and explain its relevance to soccer performance. The revised text now reads: “The Yo-Yo Intermittent Recovery Test Level 1 is a field-based assessment designed to evaluate an athlete’s ability to repeatedly perform intense exercise bouts with brief recovery periods, a key physical demand in soccer. The test consists of repeated 20-meter shuttle runs at progressively increasing speeds, interspersed with short recovery intervals, and is widely used to assess players’ aerobic and anaerobic capacity, as well as their ability to sustain high-intensity efforts throughout a match.”
#32 Beta-alanine: Are there only 2 studies to support its use in soccer players? How many studies, as referred to in lines 276-277, report no effects in soccer players? Perhaps beta-alanine should not yet be recommended for soccer players if there is not more evidence to support its use.
Response: Thank you for this important comment. You are correct that the current evidence base for beta-alanine supplementation specifically in soccer players is limited. To date, only a small number of studies have reported ergogenic effects, and several others have found no significant performance benefits in soccer-specific contexts. We agree that this limited and inconsistent evidence base warrants caution in making strong recommendations. To reflect this, we have revised the manuscript text to better contextualize the evidence and include a note of caution: “Evidence for beta-alanine supplementation in soccer players remains limited and inconsistent. While a few studies have reported improvements in high-intensity performance and fatigue resistance, several others have shown no significant effects on soccer-specific outcomes. Given the small number of studies and mixed findings, beta-alanine should be considered an emerging supplement with potential, but it cannot yet be strongly recommended for routine use in soccer without further research.” This revision acknowledges the current state of the evidence and aligns the strength of the recommendation with the quality and quantity of available data.
#33 Lines 338-340: Is there research to support this notion? Could it help recovery more during the game?
Response: Thank you for this insightful comment. There is evidence to suggest that post-exercise creatine supplementation — particularly when co-ingested with carbohydrates — can enhance muscle glycogen resynthesis (e.g., Greenhaff et al., 1994; Miny et al., 2017). This effect may be especially beneficial for recovery between matches or training sessions during periods of dense competition. However, current research does not support a significant acute, in-game recovery benefit from creatine taken immediately before or during a match. Its primary utility lies in chronic supplementation to increase intramuscular phosphocreatine stores and in post-exercise dosing to support recovery and glycogen replenishment ahead of subsequent sessions. To reflect this more accurately, we revised the text as follows: “Creatine may also enhance post-exercise glycogen resynthesis (Miny et al., 2017), and for this reason, supplementation is often recommended after matches to maximize its effects on glycogen replenishment. This strategy is particularly relevant for recovery between matches or training sessions, especially during periods of dense competition, but current evidence does not support a significant acute benefit during match play itself, as creatine’s primary ergogenic effects are linked to chronic supplementation and recovery support rather than in-game performance.”
#34 Creatine: Discuss the need to increase water consumption so that athletes do not see negative consequences.
Response: You are correct that creatine supplementation increases intracellular water retention as phosphocreatine stores expand, which can slightly increase total body water. While this is typically a beneficial adaptation, supporting muscle volume, thermoregulation, and cellular function, it also underscores the importance of maintaining adequate hydration. We have now added a clarifying sentence to the manuscript to emphasize this point: “Because creatine supplementation increases intracellular water retention, athletes should ensure adequate daily fluid intake to support proper hydration and minimize the risk of gastrointestinal discomfort, cramping, or heat-related issues during training and competition.”
#35 Creatine: As with caffeine, the research indicates that there are responders and non-responders to creatine. This should be discussed in this section.
Response: Thank you for this insightful comment. We agree that individual variability in response to creatine supplementation is an important consideration that should be addressed. Research indicates that while many athletes experience significant improvements in muscle phosphocreatine content, strength, and performance, a subset of individuals exhibit little to no measurable benefit. To address this, we have added the following text to the manuscript: “It is important to note that, similar to caffeine, there is considerable inter-individual variability in response to creatine supplementation. While many athletes experience significant increases in intramuscular phosphocreatine stores and subsequent performance benefits, others may exhibit minimal changes. Responsiveness is influenced by factors such as baseline muscle creatine content, muscle fiber composition (with a greater proportion of type II fibers often associated with a stronger response), habitual dietary intake, and training status (Kreider et al., 2017). In practice, practitioners can often identify non-responders during the initial loading phase, as these individuals typically do not show the expected increase in body mass associated with intracellular water retention. Because creatine is co-transported with sodium into muscle cells, a lack of weight gain during loading suggests insufficient intramuscular creatine accumulation to meaningfully impact performance.”
#36 There are 2 sections labelled as "2.7".
Response: Thank you for catching this, This has been corrected.
#37 Lines 416-417: Specify what is meant by "players" here. Is this specific to soccer players? Is this in athletes in general?
Response: Thank you for this helpful comment. Upon reviewing the original reference (Kahle et al., 2013), we confirmed that the study was conducted in healthy, recreationally active adults rather than soccer players specifically. We have revised the sentence to more accurately reflect the study population. It now reads: “Overall, while smaller amounts of sodium bicarbonate are generally well tolerated, some recreationally active individuals and athletes report gastrointestinal discomfort, including diarrhea, cramps, and bloating (Kahle et al., 2013).”
#38 Lines 424-427: Is it effective for sports performance? Is it well-tolerated?
Response: Thank you for this valuable comment. Current research suggests that sodium bicarbonate remains an effective ergogenic aid for high-intensity and repeated-bout exercise by enhancing extracellular buffering capacity and delaying the onset of fatigue. However, its use has traditionally been limited by gastrointestinal side effects, particularly when consumed in large doses or without proper dose-splitting strategies. Recent work on hydrogel-based sodium bicarbonate formulations (Gough et al., 2024) shows promising results in addressing this limitation. These formulations encapsulate sodium bicarbonate in a protective matrix, reducing direct exposure to gastric acid and thereby improving gastrointestinal tolerance. Early evidence also indicates that performance benefits are maintained, or in some cases enhanced, due to improved compliance and higher tolerated doses. To reflect this in the manuscript, we revised the sentence as follows: “A recent hydrogel-based formulation of sodium bicarbonate, designed to minimize gastrointestinal discomfort and reduce the need for split dosing, has shown promising results. By protecting sodium bicarbonate from stomach acid, it improves gastrointestinal tolerance and ease of ingestion while maintaining its ergogenic potential in high-intensity and repeated-exercise performance (Gough et al., 2024).”
#39 Sodium bicarbonate: Is there only one study that looks at the effects of sodium bicarbonate alone in soccer players?
Response: The body of research specifically examining sodium bicarbonate supplementation in soccer players is limited. To our knowledge, there is only one study that has investigated the effects of sodium bicarbonate alone in this population — conducted in elite Polish soccer players over an 11-day period (Zajac et al., 2018). Most other available studies, including the work in Korean elite players (Jung et al., 2021), have evaluated sodium bicarbonate in combination with other ergogenic aids (e.g., creatine), making it difficult to isolate its independent effects.
#40 Lines 434-436: Specify if this is in the context of soccer alone or if it is in all athletic context.
#41 Nitrate: Is there only one study to look at the effects of nitrate in soccer players?
Response: Thank you for this important comment. We have expanded the nitrate section to clarify that the evidence base extends beyond a single study and now includes several investigations in professional and semi-professional soccer players. For example, Nyakayiru et al. (2017) demonstrated performance improvements in Dutch professional players, while a study in Slovenian first-division players showed benefits when baseline dietary nitrate intake was below ~300 mg/day. Additionally, supplementation with L-citrulline DL-malate combined with nitrates from amaranth increased maximal sprint speed in female players from the Spanish first and second divisions. We have also incorporated findings from a recent survey showing that approximately 50% of elite nutrition practitioners recommend nitrate supplementation, highlighting its perceived utility despite heterogeneous research outcomes. These revisions strengthen the discussion by emphasizing that, although the current evidence base is still relatively limited and context-dependent, there is support for nitrate supplementation in soccer-specific settings beyond a single study.
#42 Line 455: "Yo-Yo IR1" --> More needs to be explained about this test.
Response: We have revised the manuscript to include a brief explanation of the Yo-Yo Intermittent Recovery Test Level 1 (Yo-Yo IR1), highlighting its relevance as a soccer-specific assessment of repeated high-intensity exercise capacity and its strong association with in-game performance.
#43 Line 459: Do you mean "soccer players" here?
Response: Yes. This has been corrected.
#44 "Scientific debate persists regarding the seemingly conflicting results between field studies and laboratory research on the impact of hypohydration on exercise performance." --> Explain what you mean here.
Response: We agree that the original sentence was vague and did not sufficiently explain the source of the mixed findings in the literature. We have revised the text to clarify the nature of the scientific debate and to more accurately describe the differences between laboratory and real-world research. We also replaced the term “field settings” with “applied studies” to reflect the terminology more commonly used in sports science. The revised section now reads: “Scientific debate persists because laboratory studies, which typically involve controlled fluid restriction and standardized exercise protocols, consistently show that hypohydration exceeding ~2% of body mass impairs performance, especially during endurance exercise in hot environments. By contrast, applied studies conducted under real-world competitive conditions sometimes report smaller or inconsistent effects, likely due to behavioral and motivational compensations by athletes or the ability to self-regulate exercise intensity.”
#45 Lines 477-479: This sentence is incomplete. Was the full thought included
Response: Thank you for pointing this out. You are correct that the original sentence was incomplete. We have revised it to form a complete thought and improve clarity. The updated version now reads:
“While even a mild level of hypohydration, as little as a 1% loss of body mass, can impair cognitive function, affecting task performance, reaction time, short-term memory, and mood (van Rosendal et al., 2012), these effects may further compound the physical performance decrements associated with greater fluid losses.”
#46 Glycerol: Are there any studies in soccer players at all to justify its use in the sport? Are there potential negative effects of hyperhydration or the weight gain associated with it? You mention the addition of sodium; could the same effects be accomplished with sodium ingestion alone (instead of in combination with glycerol)?
Response: Thank you for these valuable comments. We have substantially revised the Glycerol section to address your concerns. Specifically, we now: Clarify the evidence base. We explicitly state that most studies investigating glycerol-induced hyperhydration were conducted in non-soccer populations, and that direct evidence in soccer players is limited. The section now emphasizes that any application of glycerol in soccer should be considered context-dependent (e.g., hot environments, extended matches, or congested schedules) and tested during training before competitive use. Discuss potential drawbacks – We added details on the possible adverse effects of glycerol supplementation, including transient increases in body mass, gastrointestinal discomfort, and the risk of dilutional hyponatremia if fluid-electrolyte balance is not well managed. Compare with sodium alone – We expanded the discussion to address whether similar effects could be achieved with sodium ingestion alone. The revised text now notes that sodium alone is a practical and effective strategy to enhance fluid retention and plasma volume, and that the incremental benefit of adding glycerol is uncertain in soccer-specific contexts. Highlight practical recommendations. The revised text also advises practitioners to individualize hydration strategies and carefully monitor tolerance, body mass, and electrolyte balance when considering glycerol use. These additions strengthen the section by acknowledging evidence limitations, clarifying risks and alternatives, and providing more practical guidance for real-world application.
#47 Lines 516-521: Was this through supplementation or milk consumption?
Response: Thank you for pointing out the ambiguity. We have clarified in the revised manuscript that the observed ergogenic effects were reported in participants receiving milk protein supplementation, rather than from dietary milk consumption.
#48 Protein: Are there recommended ranges of consumption? How much should come through supplements vs. normal foods? Are supplements as well absorbed in the body as whole foods? Are there negative consequences of consuming too much protein? Are certain protein supplements more effective?
Response: Thank you for the helpful suggestions. We have revised the Protein section to include recommended intake ranges (1.4–2.0 g/kg/day, up to 2.2 g/kg during intense training), clarified the roles of whole foods vs. supplements, and noted that supplements are well absorbed and highly bioavailable. We also now discuss potential risks of excessive intake (>2.5–3.0 g/kg/day), including the displacement of other essential macronutrients and the possibility of increased renal stress in individuals with pre-existing kidney conditions. Additionally, we summarize differences between protein types, highlighting whey’s rapid anabolic effects, casein’s sustained release, and plant-based options. These changes provide clearer, more practical guidance on protein use in soccer.
#49 Lines 537-539: They suggested it? Or, their research supported it
Response: We have revised the sentence to clarify that Bell et al.’s findings supported the efficacy of Montmorency tart cherry concentrate in enhancing recovery rather than merely suggesting its potential.
#50 Lines 555-556: "first men team elite of the first league" --> What does this mean?
Response: Thank you for your comment. We have revised the sentence to clarify that eligibility required practitioners to be working with a elite-level men’s team competing in the highest professional division.
#51 "75% worked with soccer players on a daily basis, while 25% engaged with 580 players once a week." --> Daily vs. once per week? It seems unusual that there was nothing in between.
Response: Thank you for your observation. We have clarified this sentence to indicate that while 75% of practitioners worked with players daily, the remaining 25% engaged with them less frequently, most commonly once per week or during planned training sessions.
#52 Lines 583-584: Is this the same or different from a coach?
Response: We have clarified that the head of performance is a role distinct from a coach, typically responsible for coordinating input from nutrition, medical, strength and conditioning, and sport science staff to optimize player preparation and performance.
#53 Lines 582-584: This information is redundant with what is presented in Figure 2.
Response: Thank you for the comment. We agree that the information presented here is also displayed in Figure 2. However, we chose to retain it in the text to enhance clarity and ease of reading, allowing readers to quickly grasp key participant characteristics without needing to refer back to the figure.
#54 "Almost all participants reported an interest in and use of performance supplements." --> Separate this out because these are two separate questions.
Response: We have revised the sentence to clearly separate interest in performance supplements from their actual use, reflecting that these were distinct survey questions.
#55 "In contrast, glycerol (10.5%) was not commonly recommended (<20%)." --> Why are there 2 statistics for this line?
Response: Thank you for pointing this out. We have removed the redundant “<20%” reference and now report only the specific percentage (10.5%) to avoid confusion.
#56 Lines 589-596: This is redundant with what is displayed in Figure 3.
Response: Thank you for the comment. We agree that the information presented here is also displayed in Figure 3. However, we chose to retain it in the text to enhance clarity and ease of reading, allowing readers to quickly grasp key participant characteristics without needing to refer back to the figure.
#57 "The majority of practitioners recommend protein after exercise (85%) in combination with carbohydrates and creatine." --> Clarify if they recommended protein generally, such as what is found in whole foods, or if they recommended protein supplements specifically.
Response: We have clarified that this refers specifically to the recommendation of protein supplements post-exercise, rather than general protein intake from whole foods.
#58 Lines 601-602: Provided by who?
Response: Thank you for pointing this out. We have clarified that creatine monohydrate was the form recommended and supplied by practitioners to the players.
#59 Do you know that players were only consuming that which was provided to them by practitioners?
Response: Thank you for the comment. We have assessed dietary supplement intake only based on practitioner recommendations. It is unknown whether players consumed additional dietary ingredients independently, as this was not investigated in our study.
#60 How were nitrates administered?
Response: We have added the missing information as follows: “most commonly administered in the form of beetroot juice or concentrated nitrate beverages.”
#61 Lines 616-623: The supplementation usage seems to corroborate the evidence. Explain the discrepancies that you allude to.
Response: Thank you for the comment. We have expanded this section to explain the potential reasons for the discrepancies between research evidence and applied practice, including knowledge translation gaps, limited dissemination of evidence to practitioners, and unfamiliarity with newer findings, in addition to practical factors like tolerance, preferences, and cultural influences.
#62 "These differences underscore the influence of factors such as gastrointestinal tolerance, individual preferences of players and cultural beliefs." --> How do your findings underscore the influence of GI issues, etc.?
Response: Thank you for the comment. We have clarified how our findings support this statement by linking it directly to survey responses, which identified gastrointestinal issues, player preferences, cultural attitudes, and logistical barriers as primary reasons why certain supplements were not widely adopted despite supporting evidence.
#63 "By contrast, sodium bicarbonate, despite meta-analyses showing benefits for repeated high-intensity performance, was recommended by only 24% of practitioners, primarily due to gastrointestinal side effects and challenges with timing protocols." --> Please specific if practitioners indicated that this was the reason for low recommendations. One very important point that seems to be neglected is the common problem that happens where research is conducted but not disseminated to those who might be most likely to use it. This needs to be discussed and considered in the current findings.
Response: Thank you for this insightful comment. You are correct that our data do not directly demonstrate that gastrointestinal discomfort or timing challenges were the primary reasons practitioners avoided recommending sodium bicarbonate, those explanations were inferred from existing literature rather than explicitly stated by respondents. We have revised the text to reflect this more accurately and to incorporate the important point you raised about research dissemination. Revised sentence: “…was recommended by only 24% of practitioners. While gastrointestinal side effects and the complexity of timing protocols are frequently cited in the literature as barriers to its use, our findings also highlight a broader challenge, that evidence from research is not always effectively disseminated to or integrated by practitioners, which may contribute to the low adoption of this supplement in applied settings..”
#64 "These findings reveal that while scientific evidence provides a foundation for supplement efficacy and relation with soccer performance, translation into elite soccer requires adaptation to the complexities of the dressing room." --> Do the findings show this? Or, do they simply suggest that practitioners are not informed about all of these supplements?
Response: We agree that the original sentence overstated the findings and could be misinterpreted. Our results do not demonstrate that practitioners actively adapt evidence to the “complexities of the dressing room”; rather, they suggest that many practitioners may lack awareness of the full range of evidence-based supplements or face barriers to implementation in practice. To reflect this more accurately, we have revised the sentence as follows: “These findings suggest that, while scientific evidence provides a foundation for supplement efficacy and its relation to soccer performance, the translation of this evidence into elite soccer settings is often limited — potentially due to gaps in practitioner knowledge, lack of familiarity with certain supplements, or practical barriers to implementation.”
#65 Line 694: Clarify what is meant by a "standardized dose" here. Is this supplemented? If so, what is the formulation?
Response: Thank you for this helpful comment. We have clarified the wording to specify that the comparison refers to a supplemental form rather than an undefined “standardized dose.” We have removed standardized from the revised manuscript to simply state: “equivalent to just 800 mg of beta-alanine”
#66 Line 695: What is mean by "efficient dose" here? Do you mean "effective"?
Response: We agree that “efficient dose” was not the most accurate term in this context. We intended to refer to a dose that produces the desired physiological or performance effect. We have therefore revised the text to use the term “effective dose” instead, which more accurately reflects the intended meaning.
#67 Lines 708-712: How does this compare to their cooked counterparts?
Response: Thank you for this helpful comment. We have clarified this point in the manuscript. Cooking meat — particularly methods involving high heat or prolonged cooking times — reduces its creatine content by approximately 10–30%, primarily due to the conversion of creatine to creatinine during the heating process (Balsom et al., 1994). As a result, the actual creatine intake from cooked meat is lower than from raw meat, making it even more impractical to achieve performance-relevant doses through diet alone. We have therefore revised the text to include the following clarification: “Cooking meat typically reduces creatine content by approximately 10–30% due to the conversion of creatine to creatinine during the heating process. This further increases the amount of food required to achieve ergogenic doses, reinforcing why creatine monohydrate supplementation is generally the preferred approach.”
#68 Line 714: Define WADA
Response: This has been changed to World Anti-Doping Agency (WADA)
Reviewer 2 Report
Comments and Suggestions for Authors
Thank you for your work on the review titled “From Science to Dressing Room: Dietary Supplements for Elite Soccer Performance.” The integration of scientific evidence with practical insights from elite soccer environments is both timely and valuable. Your effort to bridge the gap between research and real-world application is commendable and contributes meaningfully to the field of sports nutrition and performance. However, some concerns should be raised and addressed to improve the paper further.
I- Abstract
- The sentence structure is awkward (“The aims of this review is…” should be “The aim of this review is…”). Also, the phrase “commonly used supplements” could be more specific—e.g., naming key categories like ergogenic aids or recovery supplements.
- Methods: The description of the questionnaire lacks detail. Who were the respondents? What was the sample size? How was the questionnaire validated?
- Results: The presentation is dense and could benefit from clearer grouping (e.g., performance enhancers vs recovery aids). Also, terms like “enhance extracellular buffering capacity” may be too technical for general readers—consider simplifying or briefly explaining.
- Results: The phrase “complexities of the dressing room” is vague. It would be stronger to specify examples like cultural resistance, taste preferences, or side effects. Also, the conclusion could better emphasize the need for education or tailored implementation strategies.
II- Introduction
- The introduction attempts to establish the importance of nutrition and supplementation in elite soccer, but it relies heavily on broad claims and quotations (e.g., Wenger’s statement) without clearly linking them to the research question. While the quote adds authority, it could be more effectively integrated to support a specific argument. Additionally, the sentence structure is occasionally awkward and repetitive—for example, listing “highly trained, elite, international-level and world-class” players is excessive and could be condensed for clarity.
- While the aim of bridging scientific evidence with practical application is commendable, the introduction lacks specificity regarding the types of supplements or performance metrics being addressed. It would benefit from a clearer definition of what constitutes “soccer performance” and how practitioner strategies will be evaluated—setting a stronger foundation for the narrative review.
III-Conclusion
The conclusion effectively summarizes the key findings, emphasizing the gap between scientific evidence and real-world application in elite soccer. It rightly highlights that supplements like caffeine, creatine, and protein are widely adopted, while others with emerging evidence are less integrated. However, the section could benefit from a more concise and structured summary. The list of supplements and their adoption rates feels repetitive and could be better grouped (e.g., widely used vs. underused) to improve clarity and impact.
Additionally, the call for future research is appropriate but could be more specific (e.g., “longitudinal studies on supplement adherence in elite soccer environments”).
Author Response
The authors are grateful to the reviewers for their thorough and constructive feedback, which has been invaluable in refining the manuscript. The suggested changes have significantly enhanced the clarity, scientific rigor, and overall quality of the work.
We would like to inform you that all changes have been highlighted in Blue.
The authors are grateful to the reviewers for their thorough and constructive feedback, which has been invaluable in refining the manuscript. The suggested changes have significantly enhanced the clarity, scientific rigor, and overall quality of the work.
- Abstract
#1 The sentence structure is awkward (‘The aims of this review is…’ should be ‘The aim of this review is…’). Also, the phrase ‘commonly used supplements’ could be more specific — e.g., naming key categories like ergogenic aids or recovery supplements.
Response: We appreciate the reviewer’s feedback. The sentence structure has been corrected, and the abstract now reads: “The aim of this review is to provide an overview of the effects of commonly used dietary supplements on soccer performance and to bridge the gap between scientific evidence and their practical application by practitioners working with elite soccer players.” While we opted to keep the phrase “commonly used dietary supplements” for conciseness in the abstract, the main manuscript text now explicitly discusses categories such as ergogenic aids, hydration strategies, and recovery-support supplements to provide more specificity.
#2 Methods: The description of the questionnaire lacks detail. Who were the respondents? What was the sample size? How was the questionnaire validated?
Response: We thank the reviewer for this helpful suggestion and have substantially improved the methodological detail in the Methods section. The revised abstract now specifies that: Insights were gathered from a cross-sectional online questionnaire completed by nutritionists, physicians, sport scientists, strength and conditioning coaches, and heads of performance working with first-division men’s teams across five European leagues. Eligible respondents were over 18 years old with more than two years of experience in elite sport. The 20-question survey, designed on Qualtrics, was pilot-tested for content validity and focused on practitioner background, beliefs about supplementation, and real-world practices. The study received ethical approval from the Ethical Independent Committee in Genoa, Italy (Ref. 2020/12). These changes provide clarity on the study population, eligibility criteria, questionnaire content, and validation approach, addressing the reviewer’s concerns while maintaining brevity.
#3 Results: The presentation is dense and could benefit from clearer grouping (e.g., performance enhancers vs recovery aids). Also, terms like “enhance extracellular buffering capacity” may be too technical for general readers—consider simplifying or briefly explaining.
Response: We agree and have reorganized the Results section to group supplements by their primary function, improving readability and interpretation: Performance enhancers: caffeine, creatine, beta-alanine, and sodium bicarbonate (with simplified explanation of their role in reducing acidity during repeated high-intensity exercise). Hydration and endurance support: dietary nitrates and glycerol. Recovery aids: protein and tart cherry. Technical language has been simplified or briefly explained (e.g., “enhance extracellular buffering capacity” is now described as “help reduce the buildup of acidity in muscles”). This restructuring improves clarity while maintaining scientific accuracy.
#4 Results: The phrase “complexities of the dressing room” is vague. It would be stronger to specify examples like cultural resistance, taste preferences, or side effects. Also, the conclusion could better emphasize the need for education or tailored implementation strategies.
Response: We fully agree and have revised the Conclusions section accordingly. The text now specifies that translation of evidence into practice is influenced by factors such as cultural resistance, taste preferences, gastrointestinal side effects, established team routines, and individual player preferences. Furthermore, we strengthened the conclusion by highlighting the importance of targeted education for players and staff, individualized supplementation plans, and close collaboration between nutritionists, coaches, and medical teams. The revised conclusion now emphasizes that tailored, multidisciplinary strategies are essential to bridge the gap between evidence and practice and maximize performance outcomes.
II- Introduction
#5 The introduction attempts to establish the importance of nutrition and supplementation in elite soccer, but it relies heavily on broad claims and quotations (e.g., Wenger’s statement) without clearly linking them to the research question. While the quote adds authority, it could be more effectively integrated to support a specific argument. Additionally, the sentence structure is occasionally awkward and repetitive—for example, listing “highly trained, elite, international-level and world-class” players is excessive and could be condensed for clarity.
We thank the reviewer for these insightful suggestions. In the revised introduction, we have reduced repetition and condensed language (e.g., “highly trained, elite, international-level, and world-class” → “elite and professional”). The Wenger quotation has been retained but reframed to explicitly support the argument that nutrition and supplementation are essential components of performance optimization strategies. The revised text now more clearly links the role of nutrition and dietary supplements to the central research question of how evidence-based strategies are implemented in elite soccer settings.
#6 While the aim of bridging scientific evidence with practical application is commendable, the introduction lacks specificity regarding the types of supplements or performance metrics being addressed. It would benefit from a clearer definition of what constitutes “soccer performance” and how practitioner strategies will be evaluated—setting a stronger foundation for the narrative review.
We fully agree with this recommendation. The revised introduction now explicitly defines “soccer performance” as a multidimensional construct encompassing physical (e.g., sprint speed, repeated high-intensity efforts, power output), technical (e.g., ball handling, passing accuracy), tactical, and cognitive components (e.g., decision-making speed and accuracy). We also specify that the review focuses on evidence-based use of ergogenic aids, recovery-support supplements, and hydration strategies, and how these are integrated into daily practice by practitioners. These clarifications provide a clearer conceptual foundation for the narrative review and strengthen the link between research evidence and applied practice.
III-Conclusion
#7 The conclusion effectively summarizes the key findings, emphasizing the gap between scientific evidence and real-world application in elite soccer. It rightly highlights that supplements like caffeine, creatine, and protein are widely adopted, while others with emerging evidence are less integrated. However, the section could benefit from a more concise and structured summary. The list of supplements and their adoption rates feels repetitive and could be better grouped (e.g., widely used vs. underused) to improve clarity and impact. Additionally, the call for future research is appropriate but could be more specific (e.g., “longitudinal studies on supplement adherence in elite soccer environments”).
We thank the reviewer for these constructive suggestions. The conclusion has been revised to improve clarity and conciseness by grouping supplements into categories (widely used vs. underused), reducing redundancy, and sharpening the key takeaways. Additionally, we have strengthened the call for future research by recommending more specific directions, such as longitudinal studies on supplement adherence, real-world implementation trials, and soccer-specific intervention protocols. These revisions improve the structure, readability, and forward-looking impact of the conclusion.
Round 2
Reviewer 2 Report
Comments and Suggestions for Authors
The paper has been significantly improved. Congratulations on your excellent work and careful revisions.